# The Surprising Effectiveness of Equivariant Models in Domains with Latent Symmetry

**Dian Wang, Jung Yeon Park, Neel Sortur, Lawson L.S. Wong, Robin Walters**[*]**, Robert Platt**[*]

Northeastern University

{wang.dian,park.jungy,sortur.n,l.wong,r.walters,r.platt}@northeastern.edu

## Abstract

Extensive work has demonstrated that equivariant neural networks can significantly improve sample efficiency and generalization by enforcing an inductive bias in the network architecture. These applications typically assume that the domain symmetry is fully described by explicit transformations of the model inputs and outputs. However, many real-life applications contain only latent or partial symmetries which cannot be easily described by simple transformations of the input. In these cases, it is necessary to *learn* symmetry in the environment instead of imposing it mathematically on the network architecture. We discover, surprisingly, that imposing equivariance constraints that do not exactly match the domain symmetry is very helpful in learning the true symmetry in the environment. We differentiate between *extrinsic* and *incorrect* symmetry constraints and show that while imposing incorrect symmetry can impede the model's performance, imposing extrinsic symmetry can actually improve performance. We demonstrate that an equivariant model can significantly outperform non-equivariant methods on domains with latent symmetries both in supervised learning and in reinforcement learning for robotic manipulation and control problems.

## 1 Introduction

Recently, equivariant learning has shown great success in various machine learning domains like trajectory prediction (Walters et al., 2020), robotics (Simeonov et al., 2022), and reinforcement learning (Wang et al., 2022c). Equivariant networks (Cohen & Welling, 2016; 2017) can improve generalization and sample efficiency during learning by encoding task symmetries directly into the model structure. However, this requires problem symmetries to be perfectly known and modeled at design time – something that is sometimes problematic. It is often the case that the designer knows that a latent symmetry is present in the problem but cannot easily express how that symmetry acts in the input space. For example, Figure 1b is a rotation of Figure 1a. However, this is not a rotation of the image – it is a rotation of the objects present in the image when they are viewed from an oblique angle. In order to model this rotational symmetry, the designer must know the viewing angle and somehow transform the data or encode projective geometry into the model. This is difficult and it makes the entire approach less attractive. In this situation, the conventional wisdom would be to discard the model structure altogether since it is not fully known and to use an unconstrained model. Instead, we explore whether it is possible to benefit from equivariant models even when the way a symmetry acts on the problem input is not precisely known. We show empirically that this is indeed the case and that an inaccurate equivariant model is often better than a completely unstructured model. For

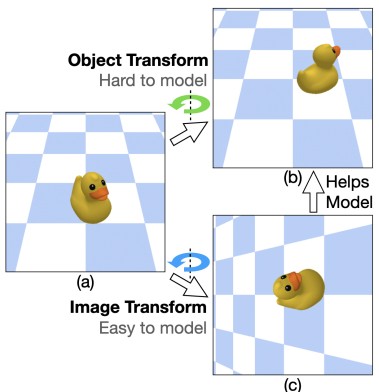

Figure 1: Object vs image transforms. Object transform rotates the object itself (b), while image transform rotates the image (c). We propose to use the image transform to help model the object transform.

---

[*]Equal Advising

example, suppose we want to model a function with the object-wise rotation symmetry expressed in Figure 1a and b. Notice that whereas it is difficult to encode the object-wise symmetry, it is easy to encode an image-wise symmetry because it involves simple image rotations. Although the image-wise symmetry model is imprecise in this situation, our experiments indicate that this imprecise model is still a much better choice than a completely unstructured model.

This paper makes three contributions. First, we define three different relationships between problem symmetry and model symmetry: *correct equivariance*, *incorrect equivariance*, and *extrinsic equivariance*. Correct equivariance means the model correctly models the problem symmetry; incorrect equivariance is when the model symmetry interferes with the problem symmetry; and extrinsic equivariance is when the model symmetry transforms the input data to out-of-distribution data. We theoretically demonstrate the upper bound performance for an incorrectly constrained equivariant model. Second, we empirically compare extrinsic and incorrect equivariance in a supervised learning task and show that a model with extrinsic equivariance can improve performance compared with an unconstrained model. Finally, we explore this idea in a reinforcement learning context and show that an extrinsically constrained model can outperform state-of-the-art conventional CNN baselines. Supplementary video and code are available at https://pointw.github.io/extrinsic_page/.

## 2 RELATED WORK

**Equivariant Neural Networks.** Equivariant networks are first introduced as G-Convolution (Cohen & Welling, 2016) and Steerable CNN (Cohen & Welling, 2017; Weiler & Cesa, 2019; Cesa et al., 2021). Equivariant learning has been applied to various types of data including images (Weiler & Cesa, 2019), spherical data (Cohen et al., 2018), point clouds (Dym & Maron, 2020), sets Maron et al. (2020), and meshes (De Haan et al., 2020), and has shown great success in tasks including molecular dynamics (Anderson et al., 2019), particle physics (Bogatskiy et al., 2020), fluid dynamics (Wang et al., 2020), trajectory prediction (Walters et al., 2020), robotics (Simeonov et al., 2022; Zhu et al., 2022; Huang et al., 2022) and reinforcement learning (Wang et al., 2021; 2022c). Compared with the prior works that assume the domain symmetry is perfectly known, this work studies the effectiveness of equivariant networks in domains with latent symmetries.

**Symmetric Representation Learning.** Since latent symmetry is not expressable as a simple transformation of the input, equivariant networks can not be used in the standard way. Thus several works have turned to learning equivariant features which can be easily transformed. Park et al. (2022) learn an encoder which maps inputs to equivariant features which can be used by downstream equivariant layers. Quessard et al. (2020), Klee et al. (2022), and Marchetti et al. (2022) map 2D image inputs to elements of various groups including $SO(3)$, allowing for disentanglement and equivariance constraints. Falorsi et al. (2018) use a homeomorphic VAE to perform the same task in an unsupervised manner. Dangovski et al. (2021) consider equivariant representations learned in a self-supervised manner using losses to encourage sensitivity or insensitivity to various symmetries. Our method may be considered as an example of symmetric representation learning which, unlike any of the above methods, uses an equivariant neural network as an encoder. Zhou et al. (2020) and Dehmamy et al. (2021) assume no prior knowledge of the structure of symmetry in the domain and learn the symmetry transformations on inputs and latent features end-to-end with the task function. In comparison, our work assumes that the latent symmetry is known but how it acts on the input is unknown.

**Sample Efficient Reinforcement Learning.** One traditional solution for improving sample efficiency is to create additional samples using data augmentation (Krizhevsky et al., 2017). Recent works discover that simple image augmentations like random crop (Laskin et al., 2020b; Yarats et al., 2022) or random shift (Yarats et al., 2021) can improve the performance of reinforcement learning. Such image augmentation can be combined with contrastive learning (Oord et al., 2018) to achieve better performance (Laskin et al., 2020a; Zhan et al., 2020). Recently, many prior works have shown that equivariant methods can achieve tremendously high sample efficiency in reinforcement learning (van der Pol et al., 2020; Mondal et al., 2020; Wang et al., 2021; 2022c), and realize on-robot reinforcement learning (Zhu et al., 2022; Wang et al., 2022a). However, recent equivariant reinforcement learning works are limited in fully equivariant domains. This paper extends the prior works by applying equivariant reinforcement learning to tasks with latent symmetries.

## 3 BACKGROUND

**Equivariant Neural Networks.** A function is equivariant if it respects symmetries of its input and output spaces. Specifically, a function $f : X \to Y$ is *equivariant* with respect to a symmetry group $G$ if it commutes with all transformations $g \in G$, $f(\rho_x(g)x) = \rho_y(g)f(x)$, where $\rho_x$ and $\rho_y$ are the *representations* of the group $G$ that define how the group element $g \in G$ acts on $x \in X$ and $y \in Y$, respectively. An equivariant function is a mathematical way of expressing that $f$ is symmetric with respect to $G$: if we evaluate $f$ for differently transformed versions of the same input, we should obtain transformed versions of the same output.

In order to use an equivariant model, we generally require the symmetry group $G$ and representation $\rho_x$ to be known at design time. For example, in a convolutional model, this can be accomplished by tying the kernel weights together so as to satisfy $K(gy) = \rho_{out}(g)K(y)\rho_{in}(g)^{-1}$, where $\rho_{in}$ and $\rho_{out}$ denote the representation of the group operator at the input and the output of the layer (Cohen et al., 2019). End-to-end equivariant models can be constructed by combining equivariant convolutional layers and equivariant activation functions. In order to leverage symmetry in this way, it is common to transform the input so that standard group representations work correctly, e.g., to transform an image to a top-down view so that image rotations correspond to object rotations.

**Equivariant SAC.** Equivariant SAC (Wang et al., 2022c) is a variation of SAC (Haarnoja et al., 2018) that constrains the actor to an equivariant function and the critic to an invariant function with respect to a group $G$. The policy is a network $\pi : S \to A \times A_\sigma$, where $A_\sigma$ is the space of action standard deviations (SAC models a stochastic policy). It defines the group action on the output space of the policy network network $\bar{a} \in A \times A_\sigma$ as: $g\bar{a} = g(a_{\text{equiv}}, a_{\text{inv}}, a_\sigma) = (\rho_{\text{equiv}}(g)a_{\text{equiv}}, a_{\text{inv}}, a_\sigma)$, where $a_{\text{equiv}} \in A_{\text{equiv}}$ is the equivariant component in the action space, $a_{\text{inv}} \in A_{\text{inv}}$ is the invariant component in the action space, $a_\sigma \in A_\sigma$, $g \in G$. The actor network $\pi$ is then defined to be a mapping $s \mapsto \bar{a}$ that satisfies the following equivariance constraint: $\pi(gs) = g(\pi(s)) = g\bar{a}$. The critic is a $Q$-network $q : S \times A \to \mathbb{R}$ that satisfies an invariant constraint: $q(gs, ga) = q(s, a)$.

## 4 LEARNING SYMMETRY USING OTHER SYMMETRIES

### 4.1 MODEL SYMMETRY VERSUS TRUE SYMMETRY

This paper focuses on tasks where the way in which the symmetry group operates on the input space is unknown. In this case the ground truth function $f : X \to Y$ is equivariant with respect to a group $G$ which acts on $X$ and $Y$ by $\rho_x$ and $\rho_y$ respectively. However, *the action $\rho_x$ on the input space is not known* and may not be a simple or explicit map. Since $\rho_x$ is unknown, we cannot pursue the strategy of learning $f$ using an equivariant model class $f_\phi$ constrained by $\rho_x$. As an alternative, we propose restricting to a model class $f_\phi$ which satisfies equivariance *with respect to a different group action $\hat{\rho}_x$*, i.e., $f_\phi(\hat{\rho}_x(g)x) = \rho_y(g)f_\phi(x)$. This paper tests the hypothesis that if the model is constrained to a symmetry class $\hat{\rho}_x$ which is related to the true symmetry $\rho_x$, then it may help learn a model satisfying the true symmetry. For example, if $x$ is an image viewed from an oblique angle and $\rho_x$ is the rotation of the objects in the image, $\hat{\rho}_x$ can be the rotation of the whole image (which is different from $\rho_x$ because of the tilted view angle). Section 4.4 will describe this example in detail.

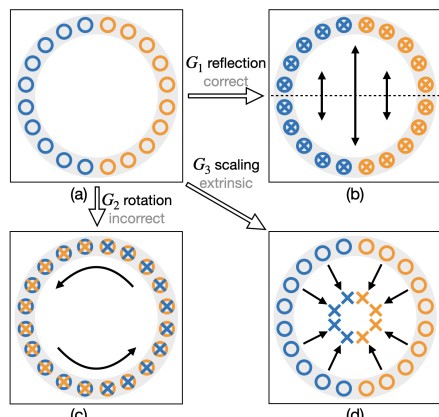

Figure 2: An example classification task for correct, incorrect, and extrinsic equivariance. The grey ring shows the input distribution. Circles are the training data in the distribution where the color shows the ground truth label. Crosses show the group transformed data.

### 4.2 CORRECT, INCORRECT, AND EXTRINSIC EQUIVARIANCE

Our findings show that the success of this strategy depends on how $\hat{\rho}_x$ relates to the ground truth function $f$ and its symmetry. We classify the model symmetry as *correct equivariance*, *incorrect*

*equivariance*, or *extrinsic equivariance* with respect to $f$. Correct symmetry means that the model symmetry correctly reflects a symmetry present in the ground truth function $f$. An extrinsic symmetry may still aid learning whereas an incorrect symmetry is necessarily detrimental to learning. We illustrate the distinction with a classification example shown in Figure 2a. (See Appendix B for a more in-depth description.) Let $D \subseteq X$ be the support of the input distribution for $f$.

**Definition 4.1.** The action $\hat{\rho}_x$ has *correct equivariance* with respect to $f$ if $\hat{\rho}_x(g)x \in D$ for all $x \in D, g \in G$ and $f(\hat{\rho}_x(g)x) = \rho_y(g)f(x)$.

That is, the model symmetry preserves the input space $D$ and $f$ is equivariant with respect to it. For example, consider the action $\hat{\rho}_x$ of the group $G_1 = C_2$ acting on $\mathbb{R}^2$ by reflection across the horizontal axis and $\rho_y = 1$, the trivial action fixing labels. Figure 2b shows the untransformed data $x \in D$ as circles along the unit circle. The transformed data $\hat{\rho}_x(g)x$ (shown as crosses) also lie on the unit circle, and hence the support $D$ is reflection invariant. Moreover, the ground truth labels $f(x)$ (shown as orange or blue) are preserved by this action.

**Definition 4.2.** The action $\hat{\rho}_x$ has *incorrect equivariance* with respect to $f$ if there exist $x \in D$ and $g \in G$ such that $\hat{\rho}_x(g)x \in D$ but $f(\hat{\rho}_x(g)x) \neq \rho_y(g)f(x)$.

In this case, the model symmetry partially preserves the input distribution, but does not correctly preserve labels. In Figure 2c, the rotation group $G_2 = \langle \mathrm{Rot}_\pi \rangle$ maps the unit circle to itself, but the transformed data does not have the correct label. Thus, constraining the model $f_\phi$ by $f_\phi(\hat{\rho}_x(g)x) = f_\phi(x)$ will force $f_\phi$ to mislabel data. In this example, for $a = \sqrt{2}/2$, $f(a, a) = $ ORANGE and $f(-a, -a) = $ BLUE, however, $f_\phi(a, a) = f_\phi(\mathrm{Rot}_\pi(a, a)) = f_\phi(-a, -a)$.

**Definition 4.3.** The action $\hat{\rho}_x$ has *extrinsic equivariance* with respect to $f$ if for $x \in D, \hat{\rho}_x(g)x \notin D$.

Extrinsic equivariance is when the equivariant constraint in the equivariant network $f_\phi$ enforces equivariance to out-of-distribution data. Since $\hat{\rho}_x(g)x \notin D$, the ground truth $f(\hat{\rho}_x(g)x)$ is undefined. An example of extrinsic equivariance is given by the scaling group $G_3$ shown in Figure 2d. For the data $x \in D$, enforcing scaling invariance $f_\phi(\hat{\rho}_x(g)x) = f_\phi(x)$ where $g \in G_3$ will not increase error, because the group transformed data (in crosses) are out of the distribution $D$ of the input data shown in the grey ring. In fact, we hypothesize that such extrinsic equivariance may even be helpful for the network to learn the ground truth function. For example, in Figure 2d, the network can learn to classify all points on the left as blue and all points on the right as orange.

### 4.3 THEORETICAL UPPER BOUND ON ACCURACY FOR INCORRECT EQUIVARIANT MODELS

Consider a classification problem over the set $X$ with finitely many classes $Y$. Let $G$ be a finite group acting on $X$. Consider a model $f_\phi : X \to Y$ with incorrect equivariance constrained to be invariant to $G$. In this case the points in a single orbit $\{gx : g \in G\}$ must all be assigned the same label $f_\phi(gx) = y$. However these points may have different ground truth labels. We classify how bad this situation is by measuring $p(x)$, the proportion of ground truth labels in the orbit of $x$ which are equal to the majority label. Let $c_p$ be the fraction of points $x \in X$ which have *consensus proportion* $p(x) = p$.

**Proposition 4.1.** *The accuracy of $f_\phi$ has upper bound* $\mathrm{acc}(f_\phi) \leq \sum_p c_p p$

See the complete version of the proposition and its proof in Appendix A. In the example in Figure 2c, we have $p \in \{0.5\}$ and $c_{0.5} = 1$, thus $\mathrm{acc}(f_\phi) \leq 0.5$. In contrast, an unconstrained model with a universal approximation property and proper hyperparameters can achieve arbitrarily good accuracy.

### 4.4 OBJECT TRANSFORMATION AND IMAGE TRANSFORMATION

In tasks with visual inputs ($X = \mathbb{R}^{c \times h \times w}$), incorrect or extrinsic equivariance will exist when the transformation of the image does not match the transformation of the latent state of the task. In such case, we call $\rho_x$ the *object transform* and $\hat{\rho}_x$ the *image transform*. For an image input $x \in X$, the image transform $\hat{\rho}_x(g)x$ is defined as a simple transformation of pixel locations (e.g., Figure 1a-c where $g = \pi/2 \in \mathrm{SO}(2)$), while the object transform $\rho_x(g)x$ is an implicit map transforming the objects in the image (e.g., Figure 1a-b where $g = \pi/2 \in \mathrm{SO}(2)$). The distinction between object transform and image transform is often caused by some symmetry-breaking factors such as camera angle, occlusion, backgrounds, and so on (e.g., Figure 1). We refer to such symmetry-breaking factors as *symmetry corruptions*.

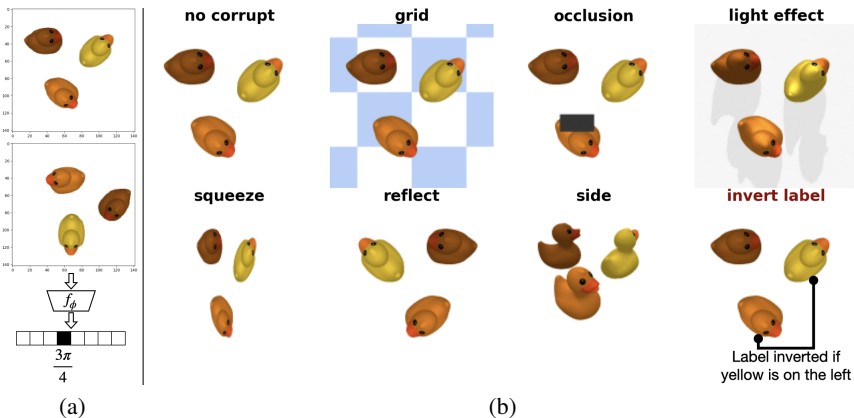

Figure 3: (a) The rotation estimation task requires the network to estimate the relative rotation between the two input states. (b) Different symmetry corruptions in the rotation estimation experiment.

## 5  EVALUATING EQUIVARIANT NETWORK WITH SYMMETRY CORRUPTIONS

Although it is preferable to use an equivariant model to enforce correct equivariance, real-world problems often contain some symmetry corruptions, such as oblique viewing angles, which mean the symmetry is latent. In this experiment, we evaluate the effect of different corruptions on an equivariant model and show that enforcing extrinsic equivariance can actually improve performance. We experiment with a simple supervised learning task where the scene contains three ducks of different colors. The data samples are pairs of images where all ducks in the first image are rotated by some $g \in C_8$ to produce the second image within each pair. Given the correct $g$, the goal is to train a network $f_\phi : \mathbb{R}^{2 \times 4 \times h \times w} \to \mathbb{R}^{|C_8|}$ to classify the rotation (Figure 3a). If we have a perfect top-down image observation, then the object transform and image transform are equal, and we can enforce the correct equivariance by modeling the ground truth function $f$ as an invariant network $f_\phi(\rho_x(g)x) = f_\phi(x)$ where $g \in \mathrm{SO}(2)$ (because the rotation of the two images will not change the relative rotation between the objects in the two images). To mimic symmetry corruptions in real-world applications, we apply seven different transformations to both pairs of images shown in Figure 3b (more corruptions are considered in Appendix E.1). In particular, for invert-label, the ground truth label $g$ is inverted to $-g$ when the yellow duck is on the left of the orange duck in the world frame in the first input image. Notice that enforcing $\mathrm{SO}(2)$-invariance in $f_\phi$ under invert-label is an incorrect equivariant constraint because a rotation on the ducks might change their relative position in the world frame and break the invariance of the task: $f(gx) \neq f(x), \exists g \in \mathrm{SO}(2)$. However, in all other corruptions, enforcing $\mathrm{SO}(2)$-invariance is an extrinsic equivariance because $gx$ will be out of the input distribution. We evaluate the equivariant network defined in group $C_8$ implemented using e2cnn (Weiler & Cesa, 2019). See Appendix D.1 for the training details.

**Comparing Equivariant Networks with CNNs.** We first compare the performance of an equivariant network (Equi) and a conventional CNN model (CNN) with a similar number of trainable parameters. The network architectures are relatively simple (see Appendix C.1) as our goal is to evaluate the performance difference between an equivariant network and an unconstrained CNN model rather than achieving the best performance in this task. In both models, we apply a random crop after sampling each data batch to improve the sample efficiency. See Appendix E.1 for the effects of random crop augmentation on learning. Figure 4 (blue vs green) shows the test accuracy of both models after convergence when trained with varying dataset sizes. For all corruptions with extrinsic equivariance constraints, the equivariant network performs better than the CNN model, especially in low data regimes. However, for invert-label which gives an incorrect equivariance constraint, the CNN outperforms the equivariant model, demonstrating that enforcing incorrect equivariance negatively impacts accuracy. In fact, based on Proposition 4.1, the equivariant network here has a theoretical upper bound performance of 62.5%. First, $p \in \{1, 0.5\}$. Then $p = 1$ when $f(x) \in \{0, \pi\} \subseteq C_8$ where $f(x) = -f(x)$ (i.e., negating the label won't change it), and $c_1 = 2/8 = 0.25$. The consensus proportion $p = 0.5$ when $f(x) \in \{\pi/4, \pi/2, 3\pi/4, 5\pi/4, 3\pi/2, 7\pi/4\} \subseteq C_8$, where half of the

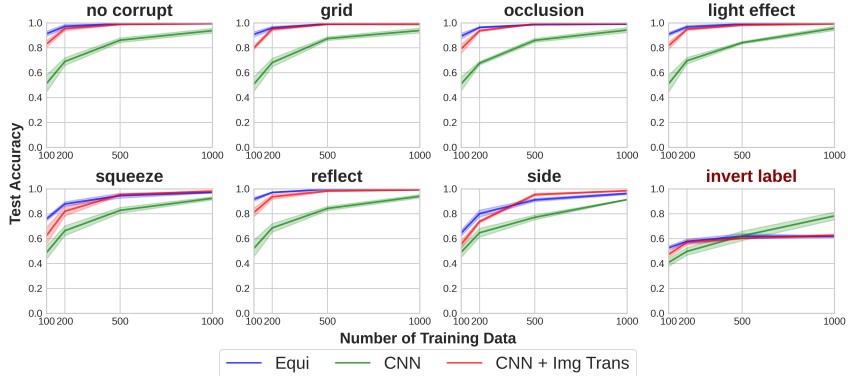

Figure 4: Comparison of an equivariant network (blue), a conventional network (green), and CNN equipped with image transformation augmentation using $C_8$ rotations (red). The plots show the prediction accuracy in the test set of the model trained with different number of training data. In all of our experiments, we take the average over four random seeds. Shading denotes standard error.

labels in the orbit of $x$ will be the negation of the labels of the other half (because half of $g \in C_8$ will change the relative position between the yellow and orange duck), thus $c_{0.5} = 6/8 = 0.75$. $\text{acc}(f_\phi) \leq 1 \times 0.25 + 0.5 \times 0.75 = 0.625$. This theoretical upper bound matches the result in Figure 4. Figure 4 suggests that even in the presence of symmetry corruptions, enforcing extrinsic equivariance can improve the sample efficiency while incorrect equivariance is detrimental.

**Extrinsic Image Augmentation Helps in Learning Correct Symmetry.** In these experiments, we further illustrate that enforcing extrinsic equivariance helps the model learn the latent equivariance of the task for in-distribution data. As an alternative to equivariant networks, we consider an older alternative for symmetry learning, data augmentation, to see whether extrinsic symmetry augmentations can improve the performance of an unconstrained CNN by helping it learn latent symmetry. Specifically, we augment each training sample with $C_8$ image rotations while keeping the validation and test set unchanged. As is shown in Figure 4, adding such extrinsic data augmentation (CNN + Img Trans, red) significantly improves the performance of CNN (green), and nearly matches the performance of the equivariant network (blue). Notice that in invert-label, adding such augmentation hurts the performance of CNN because of incorrect equivariance.

## 6  EXTRINSIC EQUIVARIANCE IN REINFORCEMENT LEARNING

The results in Section 5 suggest that enforcing extrinsic equivariance can help the model better learn the latent symmetry in the task. In this section, we apply this methodology in reinforcement learning and demonstrate that extrinsic equivariance can significantly improve sample efficiency.

### 6.1  REINFORCEMENT LEARNING IN ROBOTIC MANIPULATION

We first experiment in five robotic manipulation environments shown in Figure 6. The state space $S = \mathbb{R}^{4 \times h \times w}$ is a 4-channel RGBD image captured from a fixed camera pointed at the workspace (Figure 5). The action space $A = \mathbb{R}^5$ is the change in gripper pose $(x, y, z, \theta)$, where $\theta$ is the rotation along the $z$-axis, and the gripper open width $\lambda$. The task has latent $O(2)$ symmetry: when a rotation or reflection is applied to the poses of the gripper and the objects, the action should rotate and reflect accordingly. However, such symmetry does not exist in image space because the image perspective is skewed instead of top-down (we also perform experiments with another symmetry corruption caused by sensor occlusion in

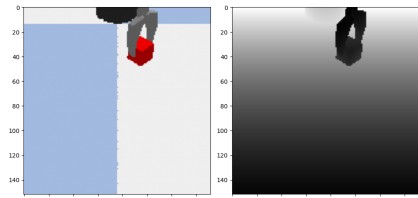

Figure 5: The image state in the Block Picking task. Left image shows the RGB channels and right image shows the depth channel.

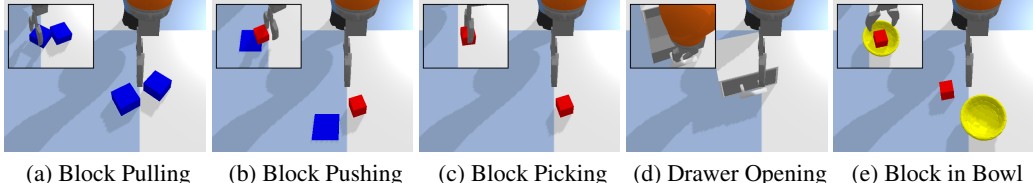

(a) Block Pulling    (b) Block Pushing    (c) Block Picking    (d) Drawer Opening    (e) Block in Bowl

Figure 6: The manipulation environments from BulletArm benchmark Wang et al. (2022b) implemented in PyBullet Coumans & Bai (2016). The top-left shows the goal for each task.

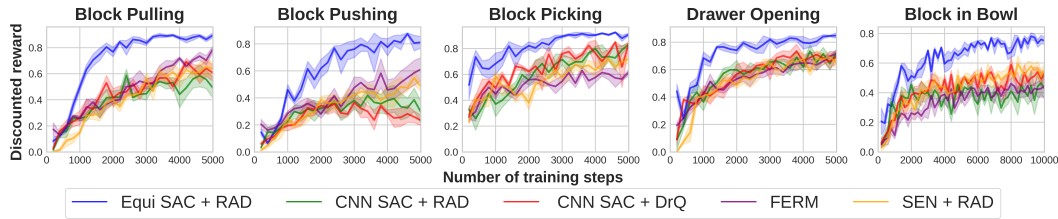

Figure 7: Comparison of Equivariant SAC (blue) with baselines. The plots show the performance of the evaluation policy. The evaluation is performed every 200 training steps.

Appendix E.3). We enforce such extrinsic symmetry (group $D_4$) using Equivariant SAC (Wang et al., 2022c;a) equipped with random crop augmentation using RAD (Laskin et al., 2020b) (Equi SAC + RAD) and compare it with the following baselines: 1) CNN SAC + RAD: same as our method but with an unconstrained CNN instead of an equivariant model; 2) CNN SAC + DrQ: same as 1), but with DrQ (Yarats et al., 2021) for the random crop augmentation; 3) FERM (Zhan et al., 2020): a combination of 1) and contrastive learning; and 4) SEN + RAD: Symmetric Embedding Network (Park et al., 2022) that uses a conventional network for the encoder and an equivariant network for the output head. All baselines are implemented such that they have a similar number of parameters as Equivariant SAC. See Appendix C.2 for the network architectures and Appendix F for the architecture hyperparameter search for the baselines. All methods use Prioritized Experience Replay (PER) (Schaul et al., 2015) with pre-loaded expert demonstrations (20 episodes for Block Pulling and Block Pushing, 50 for Block Picking and Drawer Opening, and 100 for Block in Bowl). We also add an L2 loss towards the expert action in the actor to encourage expert actions. More details about training are provided in Appendix D.2.

Figure 7 shows that Equivariant SAC (blue) outperforms all baselines. Note that the performance of Equivariant SAC in Figure 7 does not match that reported in Wang et al. (2022c) because we have a harder task setting: we do not have a top-down observation centered at the gripper position as in the prior work. Such top-down observations would not only provide correct equivariance but also help learn a translation-invariant policy. Even in the harder task setting without top-down observations, Figure 7 suggests that Equivariant SAC can still achieve higher performance compared to baselines.

## 6.2 INCREASING CORRUPTION LEVELS

In this experiment, we vary the camera angle by tilting to see how increasing the gap between the image transform and the object transform affects the performance of extrinsically equivariant networks. When the view angle is at 90 degrees (i.e., the image is top-down), the object and image transformation exactly match. As the view angle is decreased, the gap increases. Figure 8 shows the observation at 90 and 15 degree view angles. We remove the robot arm except for the gripper and the blue/white grid on the ground to remove the other symmetry-breaking components in the environment so that the camera angle

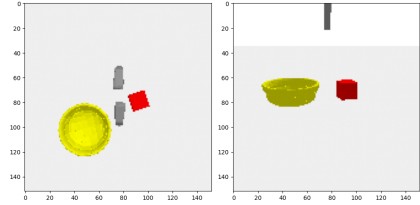

Figure 8: Left: view angle at 90 degrees. Right: view angle at 15 degrees.

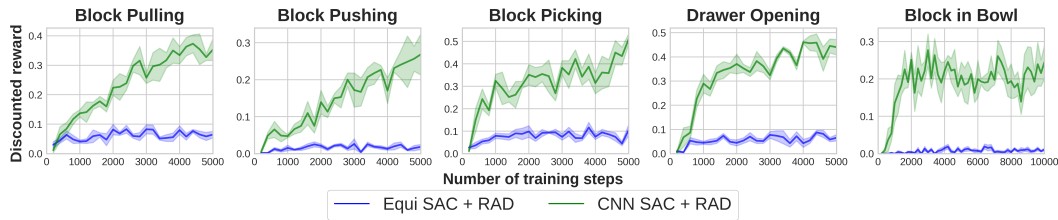

Figure 9: Comparison between Equivariant SAC (blue) and CNN SAC (green) as the view angle decreases. The plots show the evaluation performance of Equivariant SAC and CNN SAC at the end of training in different view angles.

Figure 11: Comparison between Equivariant SAC (blue) and CNN SAC (green) in an environment that will make Equivariant SAC encode incorrect equivariance. The plots show the performance of the evaluation policy. The evaluation is performed every 200 training steps.

is the only symmetry corruption. We compare Equi SAC + RAD against CNN SAC + RAD. We evaluate the performance of each method at the end of training for different view angles in Figure 9. As expected, the performance of Equivariant SAC decreases as the camera angle is decreased, especially from 30 degrees to 15 degrees. On the other hand, CNN generally has similar performance for all view angles, with the exception of Block Pulling and Block Pushing, where decreasing the view angle leads to higher performance. This may be because decreasing the view angle helps the network to better understand the height of the gripper, which is useful for pulling and pushing actions.

### 6.3 EXAMPLE OF INCORRECT EQUIVARIANCE

We demonstrate an example where incorrect equivariance can harm the performance of Equivariant SAC compared to an unconstrained model. We modify the environments so that the image state will be reflected across the vertical axis with $50\%$ probability and then also reflected across the horizontal axis with $50\%$ probability (see Figure 10). As these random reflections are contained in $D_4$, the transformed state reflect$(s), s \in S$ is affected by Equivari-

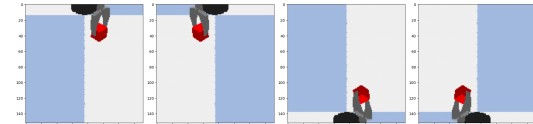

Figure 10: The environment conducts a random reflection on the state image at every step. The four images show the four possible reflections, each has 25% probability.

ant SAC's symmetry constraint. In particular, as the actor produces a transformed action for reflect when the optimal action should actually be invariant, the extrinsic equivariance constraint now becomes an incorrect equivariance for these reflected states. As shown in Figure 11, Equivariant SAC can barely learn under random reflections, while CNN can still learn a useful policy.

### 6.4 REINFORCEMENT LEARNING IN DEEPMIND CONTROL SUITE

We further apply extrinsically equivariant networks to continuous control tasks in the DeepMind Control Suite (DMC) (Tunyasuvunakool et al., 2020). We use a subset of the domains in DMC that have clear object-level symmetry and use the $D_1$ group for cartpole, cup catch, pendulum, acrobot domains, and $D_2$ for reacher domains. This leads to a total of 7 tasks, with 4 easy and 3 medium

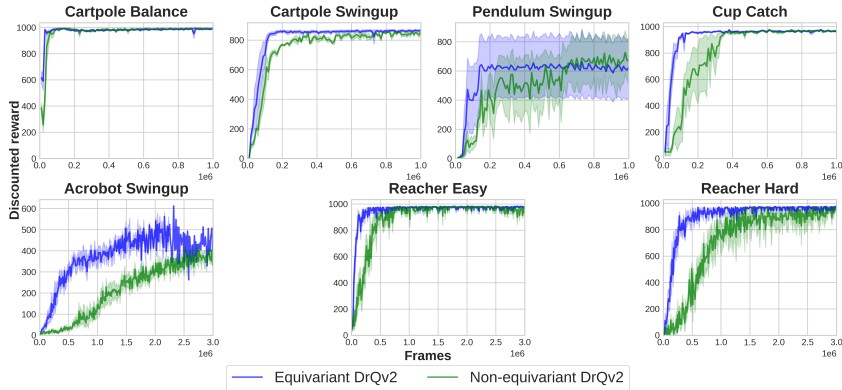

Figure 12: Comparison between Equivariant DrQv2 and Non-equivariant DrQv2 on *easy* tasks (top) and *medium* tasks (bottom). The evaluation is performed every 10000 environment steps.

level tasks as defined in (Yarats et al., 2022). Note that all of these domains are not fully equivariant as they include a checkered grid for the floor and random stars as the background.

We use DrQv2 Yarats et al. (2022), a SOTA model-free RL algorithm for image-based control, as our base RL algorithm. We create an equivariant version of DrQv2, with an equivariant actor and invariant critic with respect to the environment's symmetry group. We follow closely the architecture and training hyperparameters used in the original paper except in the image encoder, where two max-pooling layers are added to further reduce the representation dimension for faster training. Furthermore, DrQv2 uses convolution layers in the image encoder and then flattens its output to feed it into linear layers in the actor and the critic. In order to preserve this design choice for the equivariant model, we do not reduce the spatial dimensions to $1 \times 1$ by downsampling/pooling or stride as commonly done in practice. Rather we flatten the image using a process we term action restriction since the symmetry group is restricted from $\mathbb{Z}^2 \rtimes D_k$ to $D_k$. Let $I \in \mathbb{R}^{h \times w \times c}$ denote the image feature where $D_k$ acts on both the spatial domain and channels. Then we add a new axis corresponding to $D_k$ by $\tilde{I} = (gI)_{g \in D_k} \in \mathbb{R}^{h \times w \times c \times 2k}$. We then flatten to $\bar{I} = (gI)_{g \in D_k} \in \mathbb{R}^{1 \times 1 \times hwc \times 2k}$. The intermediate step $\tilde{I}$ is necessary to encode both the spatial and channel actions into a single axis which ensures the action restriction is $D_k$-equivariant. We now map back down to the original dimension with a $D_k$-equivariant $1 \times 1$ convolution. To the best of our knowledge, this is the first equivariant version of DrQv2.

We compare the equivariant vs. the non-equivariant (original) DrQv2 algorithm to evaluate whether extrinsic equivariance can still improve training in the original domains (with symmetry corruptions). In figures 12, equivariant DrQv2 consistently learns faster than the non-equivariant version on all tasks, where the performance improvement is largest on the more difficult medium tasks. In pendulum swingup, both methods have 1 failed run each, leading to a large standard error, see Figure 27 in Appendix E.4 for a plot of all runs. These results highlight that even with some symmetry corruptions, equivariant policies can outperform non-equivariant ones. See Appendix E.4.1 for an additional experiment where we vary the level of symmetry corruptions as in Section 6.2.

## 7 DISCUSSION

This paper defines correct equivariance, incorrect equivariance, and extrinsic equivariance, and identifies that enforcing extrinsic equivariance does not necessarily increase error. This paper further demonstrates experimentally that extrinsic equivariance can provide significant performance improvements in reinforcement learning. A limitation of this work is that we mainly experiment in reinforcement learning and a simple supervised setting but not in other domains where equivariant learning is widely used. The experimental results of our work suggest that an extrinsic equivariance should also be beneficial in those domains, but we leave this demonstration to future work. Another limitation is that we focus on planar equivariant networks. In future work, we are interested in evaluating extrinsic equivariance in network architectures that process different types of data.

## ACKNOWLEDGMENTS

This work is supported in part by NSF 1724257, NSF 1724191, NSF 1763878, NSF 1750649, NSF 2107256, and NASA 80NSSC19K1474. R. Walters is supported by the Roux Institute and the Harold Alfond Foundation and NSF 2134178.

## ETHIC STATEMENT

Equivariant models allow us to train robots faster and more accurately in many different tasks. Our work shows this advantage can be applied even more broadly to tasks in real-world conditions. Our method is agnostic to the morality of the actions which robots are trained for and, in that sense, can make it easier for robots to be used for either societally beneficial or detrimental tasks.

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

## A   THEORETICAL UPPER BOUND ON ACCURACY FOR MODELS WITH INCORRECT SYMMETRY

We consider a classification problem over the set $X$ with finitely many classes $Y$. Let $m = |Y|$ be the number of classes. Let $l \colon X \to Y$ be the true labels. Let $G$ be a finite group acting on $X$. We assume the action of $G$ on $X$ is density preserving. That is, if $p_X$ is the density function corresponding to the input domain, then $p_X(gx) = p_X(x)$. Denote the orbit of a point $x \in X$ by $Gx = \{gx : g \in G\}$ and the stabilizer by $G_x = \{g \in G : gx = x\}$. By the orbit-stabilizer theorem $|G| = |G_x||Gx|$.

Now consider a model $f \colon X \to Y$ with incorrect equivariance constrained to be invariant to $G$. We partition the input set into subsets $X = \coprod_{k=1}^{m} X_k$ where

$$X_k = \{x \in X : |l(Gx)| = k\}.$$

If $f$ has correct equivariance then $X = X_1$. Incorrect equivariance implies that there are orbits $Gx$ which are assigned more than one label. Since $f$ is constrained to be equivariant such orbits will necessarily result in some errors. We give an upper bound on that error. Define $c_k = \mathbb{P}(x \in X_k)$. Note that since $X_k$ give a partition, $\sum_{k=1}^{m} c_k = 1$. Also, $X_k$ is empty for $k > |G|$ since the number of labels assigned to an orbit is also upper bounded by the number of points in the orbit which is at most $|G|$. Letting $K = \min(|Y|, |G|)$, we have $X = \coprod_{k=1}^{K} X_k$.

**Proposition A.1.** *The accuracy of $f$ has upper bound* $\mathrm{acc}(f) \leq 1 - \sum_{k=1}^{K} c_k(k-1)/|G|$.

In contrast, we can choose an unconstrained model from a model class with a universal approximation property and given properly chosen hyperparameters find a model with arbitrarily good accuracy.

*Proof.* Let $y = l(x)$. Then $\mathrm{acc}(f) = \mathbb{E}_{x \in X}[\delta(f(x) = y)]$. Since the action of $G$ is density preserving, applying an element of $G$ before sampling does not affect the expectation, $\mathbb{E}_{x \in X}[\delta(f(x) = y)] = \mathbb{E}_{x \in X}[\delta(f(gx) = y)]$ and so

$$\mathrm{acc}(f) = \frac{1}{|G|} \sum_{g \in G} \mathbb{E}_{x \in X}[\delta(f(gx) = y)].$$

If we split the expectation over the partition $X = \coprod_{k=1}^{K} X_k$ we get

$$\frac{1}{|G|} \sum_{g \in G} \sum_{k=1}^{K} c_k \mathbb{E}_{x \in X_k}[\delta(f(gx) = y)].$$

Interchanging sums gives

$$\sum_{k=1}^{K} c_k \left( \mathbb{E}_{x \in X_k} \left[ \frac{1}{|G|} \sum_{g \in G} \delta(f(gx) = y) \right] \right).$$

By the orbit-stabilizer theorem,

$$\frac{1}{|G|} \sum_{g \in G} \delta(f(gx) = y) = \frac{|G_x|}{|G|} \sum_{x' \in Gx} \delta(f(x') = y) = \frac{|1|}{|Gx|} \sum_{x' \in Gx} \delta(f(x') = y)$$

which is the average accuracy over the orbit $Gx$. Since $f$ is constrained to a single value of the orbit, and $k$ different true labels appear, the highest accuracy attainable is when $|G| = |G_x|$ and the true labels are maximally unequally distributed such that 1 point in the orbit takes each of $k - 1$ labels and all the other $|G| - (k-1)$ points receive a single label. In this case accuracy can be maximized by choosing $f(x')$ to be this majority label, and

$$\frac{|1|}{|Gx|} \sum_{x' \in Gx} \delta(f(x') = y) \leq 1 - \frac{k-1}{|G|}.$$

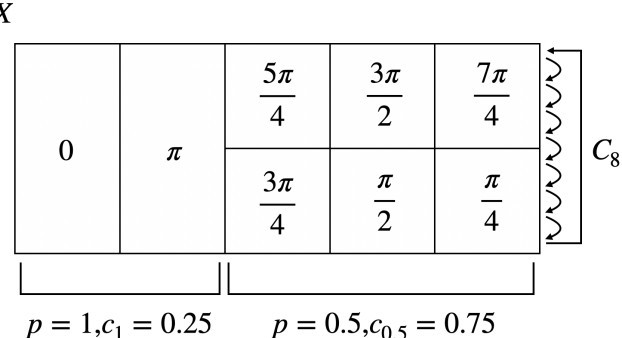

Figure 13: Demonstration of the upper bound of an equivariant model under invert label corruption in our supervised learning experiment. The number on each partition shows the ground truth label.

Substituting back in,

$$\text{acc}(f) \leq \sum_{k=1}^{K} c_k \left( \mathbb{E}_{x \in X_k} \left[ 1 - \frac{k-1}{|G|} \right] \right)$$

$$= 1 - \sum_{k=1}^{K} c_k \left( \frac{k-1}{|G|} \right)$$

since $\frac{k-1}{|G|}$ is constant over $X_k$ and $\sum_{k=1}^{K} c_k = 1$. $\qquad\square$

Note that the assumption that $|G| = |Gx|$ and that the labels on a given orbit are maximally un-equally distributed need not hold in general and thus this bound is not tight. In order to produce a tight upper bound, consider a partition $X = \coprod_p X_p$ where $X_p = \{x \in X : (\max_y |f^{-1}(y) \cap Gx|)/|Gx| = p\}$ and define $c_p = \mathbb{P}(x \in X_p)$. The set $X_p$ contains points in orbits where the majority label covers a fraction $p$ of the points. Note that although $p$ is a fraction between 0 and 1, there are only finitely many possible values of $p$ since the numerator and denominator and bounded natural numbers. We may thus sum over the values of $p$.

**Proposition A.2.** *The accuracy of $f$ has upper bound* $\text{acc}(f) \leq \sum_p c_p p$.

*Proof.* The proof is similar to the proof of Proposition A.1 replace $X_k$ and $c_k$ with $X_p$ and $c_p$ respectively. For $x \in X_p$, the term $\frac{|1|}{|Gx|} \sum_{x' \in Gx} \delta(f(x') = y)$ can be upper bounded by choosing the majority label yielding $\frac{|1|}{|Gx|} \sum_{x' \in Gx} \delta(f(x') = y) \leq p$. The bound then follows as before. $\qquad\square$

This is a tight upper bound since assigning any but the majority label would result in lower accuracy.

Figure 13 demonstrates the upper bound of an incorrectly constrained equivariant network with the invert label corruption in Section 5, where $\text{acc}(f) \leq 0.25 \times 1 + 0.75 \times 0.5 = 0.625$.

# B  CORRECT, INCORRECT, AND EXTRINSIC EQUIVARIANCE EXAMPLES

In this section, we describe how the model symmetry transforms data under correct, incorrect, and extrinsic equivariance and how such transformations relate to the true symmetry present in the task using the example of Section 4.2. The ground truth function $f : X \to Y$ is a mapping from $X = \mathbb{R}^2$ to $Y = \{\text{ORANGE}, \text{BLUE}\}$. Let $(a, a), (-a, -a), (-a, a), (b, c)$ be the coordinates of four points in the data distribution on the unit circle (Figure 14a). The ground truth labels for these points are: $f(a, a) = \text{ORANGE}, f(-a, -a) = \text{BLUE}, f(-a, a) = \text{BLUE}, f(b, c) = \text{ORANGE}$.

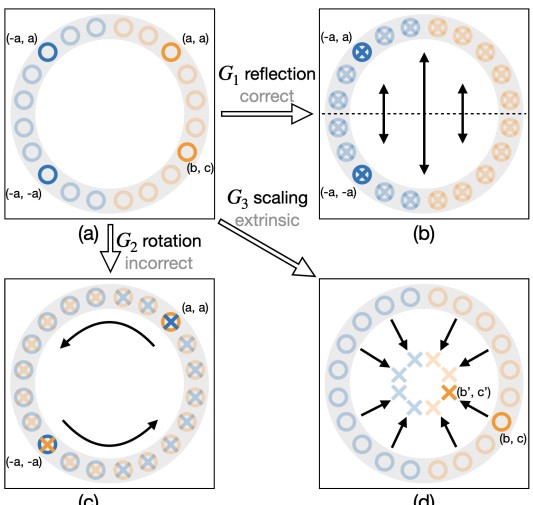

Figure 14: An example classification task for correct, incorrect, and extrinsic equivariance. The input distribution is shown as a gray ring. The training data samples are shown as circles, where the color is the ground truth label. Crosses represent the group transformed data. The opaque points highlight the example points while other points are semitransparent.

### B.1 Correct Equivariance

**Definition 4.1.** The action $\hat{\rho}_x$ has *correct equivariance* with respect to $f$ if $\hat{\rho}_x(g)x \in D$ for all $x \in D, g \in G$ and $f(\hat{\rho}_x(g)x) = \rho_y(g)f(x)$.

Consider the reflection group $G = C_2 = \{1, r\}$ (where $r$ is the reflection along the horizontal axis) acting on $X$ by $\hat{\rho}_x(1) = Id$ or $\hat{\rho}_x(r) = \left(\begin{smallmatrix} 1 & 0 \\ 0 & -1 \end{smallmatrix}\right)$ and $Y$ via $\rho_y = Id$, the trivial action fixing the labels (Figure 14b). If we define an equivariant model with respect to $\hat{\rho}_x$ and $\rho_y$, then the model's symmetry preserves the problem symmetry. For example, consider the point $(-a, a)$, $r \in G_1$ is the reflection so that $\hat{\rho}_x(r) = \left(\begin{smallmatrix} 1 & 0 \\ 0 & -1 \end{smallmatrix}\right)$ and $\hat{\rho}_x(r)(-a, a) = (-a, -a)$. Since the model $f_\phi$ is $G_1$-equivariant, $f_\phi(\hat{\rho}_x(r)x) = \rho_y(r)f_\phi(x)$. Substituting $\rho_y = Id$ and $x = (-a, a)$, we obtain $f_\phi(-a, -a) = f_\phi(-a, a)$, meaning that the output of $f_\phi(-a, -a)$ and $f_\phi(-a, a)$ are constrained to be equal. Thus the invariance property in the ground truth function $f$ where $f(-a, -a) = f(-a, a) = \text{BLUE}$ is preserved (notice that this applies to all $x \in X$). We call this correct equivariance.

### B.2 Incorrect Equivariance

**Definition 4.2.** The action $\hat{\rho}_x$ has *incorrect equivariance* with respect to $f$ if there exist $x \in D$ and $g \in G$ such that $\hat{\rho}_x(g)x \in D$ but $f(\hat{\rho}_x(g)x) \neq \rho_y(g)f(x)$.

Consider the rotation group $G_2 = \langle \text{Rot}_\pi \rangle$ (Figure 14c) which acts via $\hat{\rho}_x$ on $X$ via a rotation matrix of $\pi$ and acts on $Y$ via $\rho_y = Id$. If we define an equivariant model with respect to $\hat{\rho}_x$ and $\rho_y$, the network's symmetry will conflict with the problem's symmetry. For example, consider the point $(a, a)$ and let $g \in G_2$ be the rotation action so that $\hat{\rho}_x(g) = \left(\begin{smallmatrix} -1 & 0 \\ 0 & -1 \end{smallmatrix}\right)$ and $\hat{\rho}_x(g)(a, a) = (-a, -a)$. As the model $f_\phi$ is $G_2$-equivariant, $f_\phi(\hat{\rho}_x(g)x) = \rho_y(g)f_\phi(x)$. Substituting $\rho_y = Id$ and $x = (a, a)$, we get $f_\phi(-a, -a) = f_\phi(a, a)$. However, this constraint interferes with the ground truth function $f$ as $f(-a, -a) = \text{BLUE}$ and $f(a, a) = \text{ORANGE}$. We call this incorrect equivariance.

### B.3 Extrinsic Equivariance

**Definition 4.3.** The action $\hat{\rho}_x$ has *extrinsic equivariance* with respect to $f$ if for $x \in D$, $\hat{\rho}_x(g)x \notin D$.

Consider the scaling group $G_3$ acting on $X$ by scaling the vector and on $Y$ via $\rho_y = Id$ (Figure 14d). If we define an equivariant model with respect to $\hat{\rho}_x$ and $\rho_y$, the group-transformed data will be

outside the input distribution. Consider the point $(b, c)$ and let $g \in G_3$ be the scaling action so that $\hat{\rho}_x(g)(b, c) = (b', c')$. Since the model $f_\phi$ is $G_3$-equivariant, $f_\phi(\hat{\rho}_x(g)x) = \rho_y(g)f_\phi(x)$. Substituting $\rho_y = Id$ and $x = (b, c)$ we have $f_\phi(b', c') = f_\phi(b, c)$ meaning that the output of $f_\phi(b', c')$ and $f_\phi(b, c)$ are constrained to be equal. However, $(b', c')$ is outside of the input distribution (gray ring) and thus the ground truth $f(b', c')$ is undefined. We call this extrinsic equivariance.

Intuitively, it is easy to see in this example how extrinsic equivariance would help the model learn $f$. If the model $f_\phi$ is equivariant to the scale group $G_3$, then it can generalize to "scaled" up or down versions of the input distribution and "covers" more of the input space $\mathbb{R}^2$. As such, the model may learn the decision boundary (the vertical axis) more easily because of its equivariance compared to a non-equivariant model, even if the equivariance is extrinsic.

## C  NETWORK ARCHITECTURE

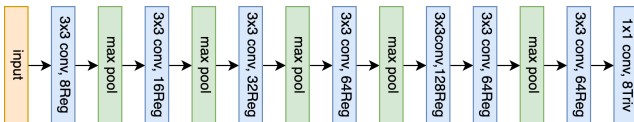

Figure 15: Network architecture of the equivariant network in the supervised learning experiment.

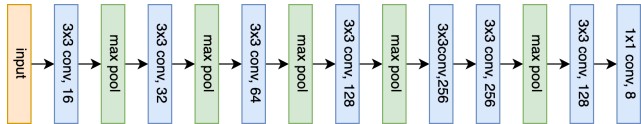

Figure 16: Network architecture of the CNN network in the supervised learning experiment.

Table 1: Number of trainable parameters of the equivariant network (Equi) and conventional CNN network (CNN) in the supervised learning task.

| Network | Equi | CNN |
|---|---|---|
| Number of Parameters | 1.11 million | 1.28 million |

### C.1  SUPERVISED LEARNING

Figure 15 shows the network architecture of the equivariant network and Figure 16 shows the network architecture of the CNN network in Section 5. Both networks are 8-layer convolutional neural networks. The equivariant network is implemented using the e2cnn (Weiler & Cesa, 2019) library, where the hidden layers are defined using the regular representation and the output layer is defined using the trivial representation. Table 1 shows the numbers of trainable parameters in both networks, where both networks have a similar number with a slight advantage in the CNN.

### C.2  REINFORCEMENT LEARNING IN ROBOTIC MANIPULATION

Figure 17 shows the network architecture of Equivariant SAC used in manipulation tasks in Section 6.1. All hidden layers are implemented using the regular representation. For the actor (top), the output is a mixed representation containing one standard representation for the $(x, y)$ actions, one signed representation for the $\theta$ action, and seven trivial representations for the $(z, \lambda)$ actions and the standard deviations of all action components. Figure 18 shows the network architecture of CNN SAC for both RAD and DrQ. Figure 19 shows the network architecture of FERM. Figure 20 shows the network architecture of SEN.

Table 2 shows the number of trainable parameters for each model. All baselines have slightly more parameters compared with Equivariant SAC.

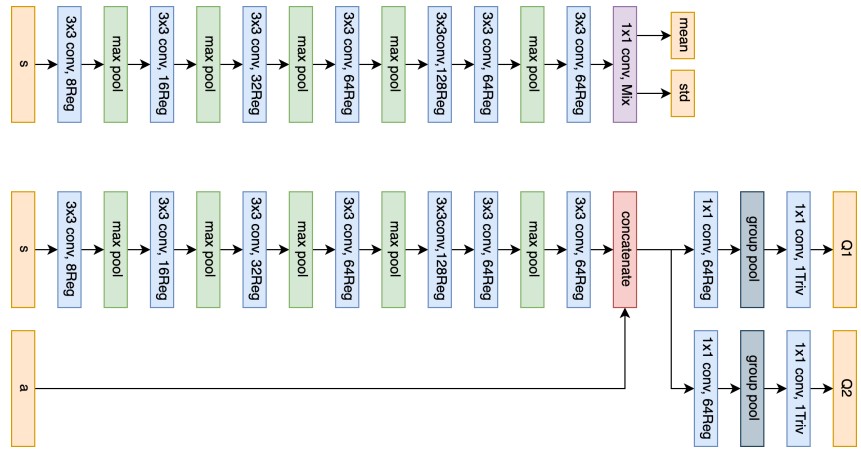

Figure 17: Network architecture of Equivariant SAC in robotic manipulation tasks.

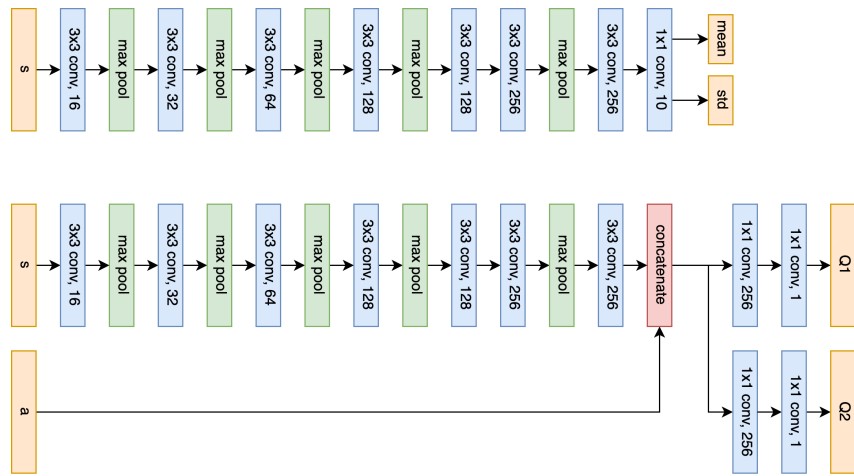

Figure 18: Network architecture of CNN SAC in robotic manipulation tasks.

# D TRAINING DETAILS

## D.1 SUPERVISED LEARNING

We implement the environment in the PyBullet simulator (Coumans & Bai, 2016). The ducks are located in a workspace with a size of $0.3m \times 0.3m$. The pixel size of the image is $152 \times 152$ (and will be cropped to $128 \times 128$ during training). We implement the training in PyTorch (Paszke et al., 2017) using a cross-entropy loss. The output of the model is the score for each $g \in C_8$. We use the Adam optimizer (Kingma & Ba, 2014) with a learning rate of $10^{-4}$. The batch size is 64. In all training, we perform a three-way data split with $N$ training data, 200 holdout validation data, and 200 holdout test data. The training is terminated either when the validation prediction success rate does not improve for 100 epochs or when the maximum epoch (1000) is reached.

## D.2 REINFORCEMENT LEARNING IN ROBOTIC MANIPULATION

We use the environments provided by the BulletArm benchmark (Wang et al., 2022b) implemented in the PyBullet simulator (Coumans & Bai, 2016). The workspace's size is $0.4m \times 0.4m \times 0.24m$. The pixel size of the image observation is $152 \times 152$ (and will be cropped to $128 \times 128$ during training). The action space is $A_x, A_y, A_z = [-0.05m, 0.05m]$ for the change of $(x, y, z)$ position of the gripper; $A_\theta = [-\frac{\pi}{4}, \frac{\pi}{4}]$ for the change of top-down rotation of the gripper; and $A_\lambda = [0, 1]$ for the open width of the gripper where 0 means fully close and 1 means fully open. All environ-

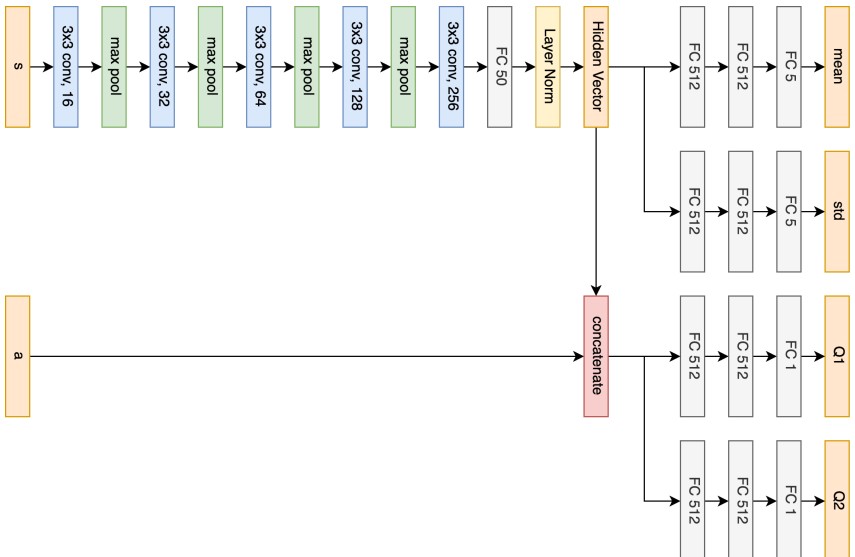

Figure 19: Network architecture of FERM in robotic manipulation tasks.

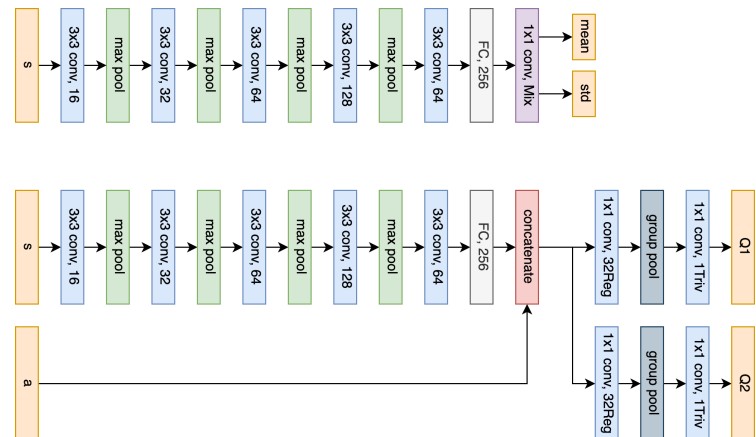

Figure 20: Network architecture of SEN in robotic manipulation tasks.

ments have a sparse reward: +1 for reaching the goal and 0 otherwise. During training, we use 5 parallel environments where a training step is performed after all 5 parallel environments perform an action step. The evaluation is performed every 200 training steps. We implement the training in PyTorch (Paszke et al., 2017). We use the Adam optimizer (Kingma & Ba, 2014) with a learning rate of $10^{-3}$. The batch size is 128. The entropy temperature for SAC is initialized at $10^{-2}$. The target entropy is $-5$. The discount factor $\gamma = 0.99$. The Prioritized Experience Replay (PER) (Schaul et al., 2015) has a capacity of 100,000 transitions with prioritized replay exponent of $\alpha = 0.6$ and prioritized importance sampling exponent $\beta_0 = 0.4$ as in Schaul et al. (2015). The expert transitions are given a priority bonus of $\epsilon_d = 1$.

The contrastive encoder of the FERM baseline has an encoding size of 50 as in Zhan et al. (2020). The FERM baseline's contrastive encoder is pre-trained for 1.6k steps using the expert data as in Zhan et al. (2020). In DrQ, the number of augmentations for calculating the target $K$ and the number of augmentations for calculating the loss $M$ are both 2 as in Yarats et al. (2021).

Table 2: Number of trainable parameters of Equivariant SAC, CNN SAC, FERM, and SEN in the reinforcement learning task in robotic manipulation. Notice that FERM has a shared encoder between the actor and the critic so the total number of parameters is smaller than the sum of the actor parameter and the critic parameter.

| Network | Equi SAC | CNN SAC | FERM | SEN |
|---|---|---|---|---|
| Number of Actor Parameters | 1.11 million | 1.13 million | 1.79 million | 1.22 million |
| Number of Critic Parameters | 1.18 million | 1.27 million | 1.90 million | 1.24 million |
| Number of Total Parameters | 2.29 million | 2.40 million | 2.34 million | 2.46 million |

### D.3 REINFORCEMENT LEARNING IN DEEPMIND CONTROL SUITE

Sample images of each environment are shown in Figure 21. Environment observations are 3 consecutive frames of RGB images of size $85 \times 85$, in order to infer velocity and acceleration. Note that we use odd-sized image sizes instead of $84 \times 84$ used in Yarats et al. (2022), as the DrQv2 architecture contains a convolutional layer with stride 2 and this breaks equivariance for even-sized spatial inputs (Mohamed et al., 2020). For each environment, an episode lasts 1000 steps where each step has a reward between 0 and 1.

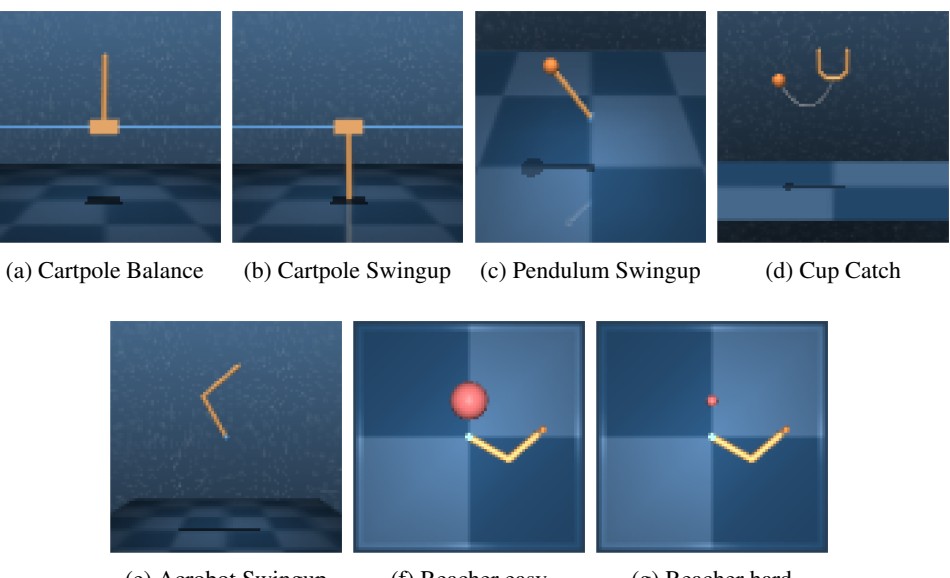

(a) Cartpole Balance    (b) Cartpole Swingup    (c) Pendulum Swingup    (d) Cup Catch

(e) Acrobot Swingup    (f) Reacher easy    (g) Reacher hard

Figure 21: DeepMind Control Suite: images of *easy* (top) and *medium* (bottom) tasks.

We modify the original DrQv2 by making the encoder map down to a smaller spatial output, leading to faster training. The second and third convolutional blocks have an added max-pooling layer, leading to a spatial output size of $7 \times 7$. As the equivariant version of DrQv2 has an additional convolutional layer after the action restriction, the non-equivariant version also has an additional convolutional layer at the end of the encoder. We also scale the number of channels by $\sqrt{|G|}$ in order to preserve roughly the same number of parameters as the non-equivariant version.

The policy is evaluated by averaging the return of 10 episodes every 10000 environment steps. In all DMC experiments, we plot the mean and the standard error over 4 seeds. All other training details and hyperparameters are kept the same as in Yarats et al. (2022).

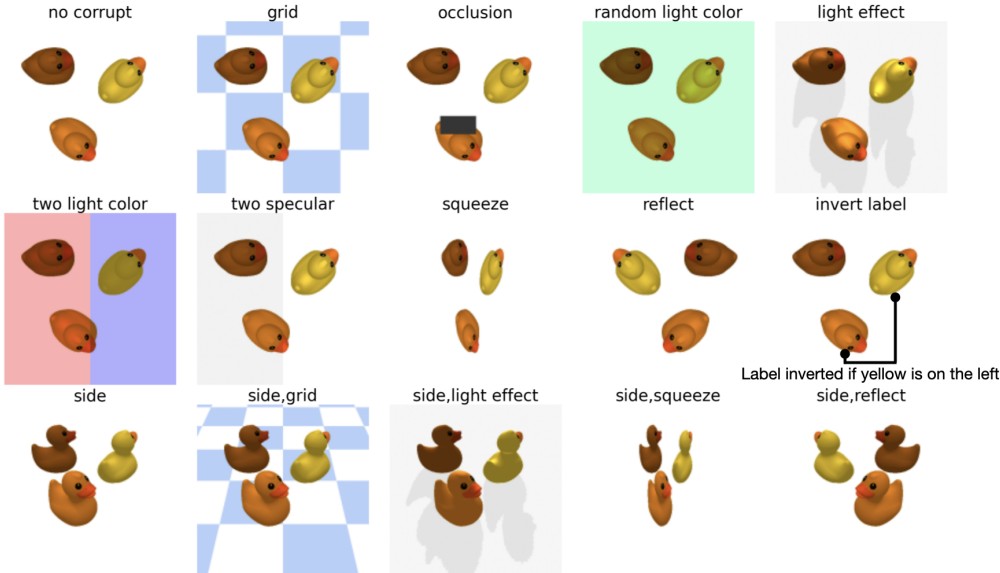

Figure 22: All symmetry corruptions in the rotation estimation experiment.

# E    ADDITIONAL EXPERIMENTS

## E.1    SUPERVISED LEARNING WITH MORE SYMMETRY CORRUPTION TYPES

In this section, we demonstrate the experiment in Section 5 in more symmetry corruptions. Figure 22 shows the 15 different corruptions. We also show the performance of 'Equi', 'CNN', and 'CNN + Img Trans' without the random crop augmentation used in Section Section 5 (labeled as 'no Crop' variations). The result is shown in Figure 23. First, comparing blue vs green, and purple vs orange, the equivariant network always outperforms the CNN with or without random crop augmentation, especially with fewer data. Second, comparing blue vs purple, and green vs orange, random crop generally helps both the equivariant network and the CNN network. Third, comparing red vs green, and cyan vs orange, adding the image transformation augmentation improves the performance of CNN. Notice that the condition reverse is an outlier because the equivariant network has incorrect equivariance, where the CNN methods (green and orange) without image transformation augmentation have the best performance.

## E.2    RL IN MANIPULATION WITHOUT RANDOM CROP

In this section, we demonstrate the performance of Equivariant SAC and CNN SAC without random crop augmentation using RAD. As is shown in Figure 24, both methods work poorly without the random crop augmentation.

## E.3    RL IN MANIPULATION WITH OCCLUSION CORRUPTION

In this section, we perform the same experiment as in Section 6.1 with a different type of symmetry corruption: occlusion due to orthographic projection using a single camera. Instead of using an RGBD image observation as in Section 6.1, we take the depth channel from the RGBD image and perform an orthographic projection at the gripper's position (Figure 25). This is the same process as in Wang et al. (2022a) to generate a top-down image for equivariant learning, however, since we only have one camera instead of two as in the prior work, this orthographic projection will have missing depth values due to occlusion and thus leads to an extrinsic equivariant constraint. Figure 26 shows the results. Similar as in Section 6.1, Equivariant SAC outperforms all baselines with a significant margin.

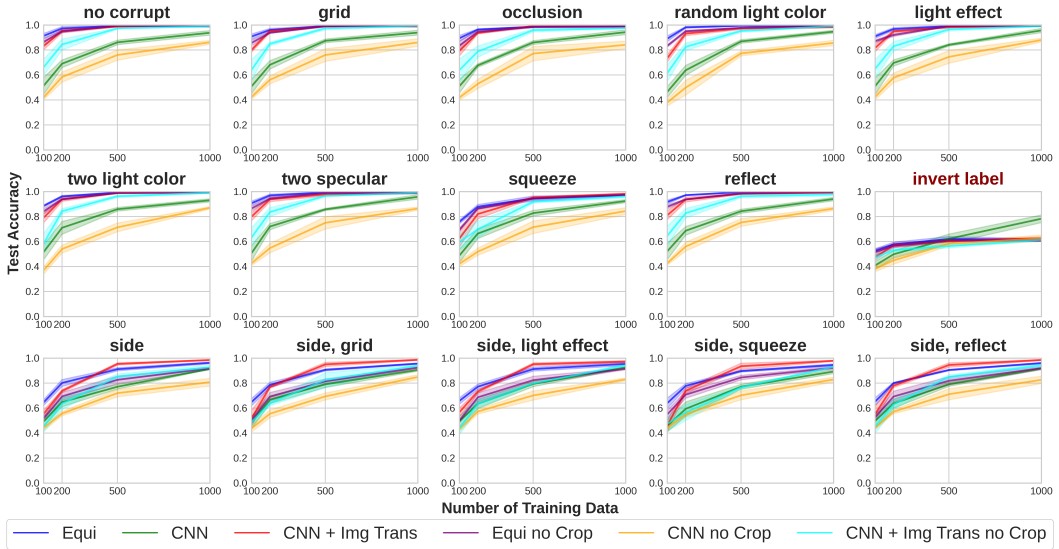

Figure 23: Comparison of an equivariant network (blue), a conventional network (green), CNN equipped with image transformation augmentation (red), and their variation without random crop augmentation (purple, orange, cyan). The plots show the prediction accuracy in the test of the model trained with different number of training data. Results are averaged over four runs. Shading denotes standard error.

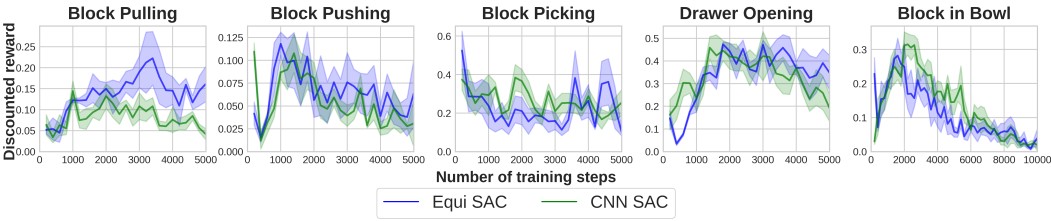

Figure 24: Comparison between Equivariant SAC and CNN SAC without data augmentation using RAD. The plots show the performance (in terms of discounted reward) of the evaluation policy. The evaluation is performed every 200 training steps. Results are averaged over four runs. Shading denotes standard error.

## E.4 RL IN DEEPMIND CONTROL SUITE

Figure 27 is another visualization of equivariant vs non-equivariant DrQv2 on the original pendulum swingup environment. As each method has 1 failed seed, we plot all runs with slightly different color shades. If we exclude the failed run from each method, it can easily be seen that equivariant DrQv2 learns faster than the non-equivariant version.

### E.4.1 INCREASING SYMMETRY CORRUPTIONS

In these experiments, we modify some domains to have different levels of symmetry-breaking corruptions. For cartpole and cup catch, we either remove the gridded floor and background (None) to make the observation perfectly equivariant or keep the floor and background and further change the camera angle by rolling ($30° − 75°$), increasing the level of corruption. For reacher, we use the same modifications but tilt the camera instead of rolling. See Figure 3 for sample images. In order to see the effects of increasing corruption on learning, we plot the mean discounted reward when both methods have converged (30k frames for cartpole and cup catch, 1.5M frames for reacher). Figure 28 shows that both the equivariant and non-equivariant DrQv2 surprisingly perform quite well across all corruption levels, with the exception of $75°$ on reacher. The equivariant policy seems to converge to a slightly higher discounted reward than the non-equivariant version, though the dif-

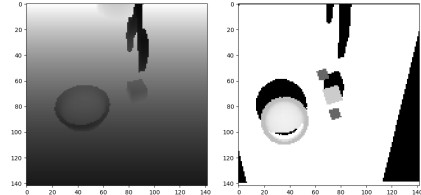

Figure 25: Left: the depth image taken from a depth camera. Right: the orthographic projection centered at the gripper position generated from the left image, where the black areas are missing depth values due to occlusions.

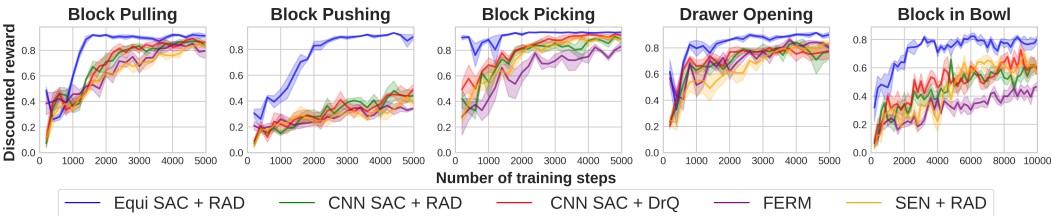

Figure 26: Comparison of Equivariant SAC (blue) with baselines in environments with occlusion corruption. The plots show the performance (in terms of discounted reward) of the evaluation policy. The evaluation is performed every 200 training steps. Results are averaged over four runs. Shading denotes standard error.

ference is not significant. On reacher, changing the camera angle may have affected both methods by making the task more difficult for both an equivariant and regular CNN encoder.

## F   BASELINE ARCHITECTURE SEARCH

### F.1   CNN SAC ARCHITECTURE SEARCH

This section demonstrates the architecture search for CNN SAC. We consider three different architectures (all with a similar amount of trainable parameters): 1) conv (Figure 18): a CNN network with the same structure as Equivariant SAC, where all layers are implemented using convolutional layers. 2) fc1 (Figure 29): a CNN network that replaces some layers in 1) with fully connected layers. 3) fc2 (Figure 30): similar as 2), but with fewer convolutional layers and more weights in the FC layer. We evaluate the three network architectures with SAC equipped random crop augmentation using RAD (Laskin et al., 2020b).

Figure 31 shows the result, where all three variations have a similar performance. We use conv in the main paper since it has a similar structure as Equivariant SAC.

### F.2   FERM ARCHITECTURE SEARCH

This section demonstrates the architecture search for FERM. We consider four different architectures: 1) sim total 1 (Figure 33) and 2) sim total 2 (Figure 19) are two different architectures with the similar amount of total trainable parameters as Equivariant SAC. 3) sim enc (Figure 32) has similar amount of trainable parameters in the encoder as Equivariant SAC's encoder. Notice that since FERM share an encoder between the actor and the critic while Equivariant SAC has separate encoders, having the similar amount of parameters in the encoder will lead to fewer total parameter in FERM compared with Equivariant SAC. 4) ferm ori (Figure 34) is the same network architecture used in the FERM paper (Zhan et al., 2020).

Figure 35 shows the comparison across the four architectures. 'sim total 2' has a marginal advantage compared with the other three variations, so we use it in the main paper.

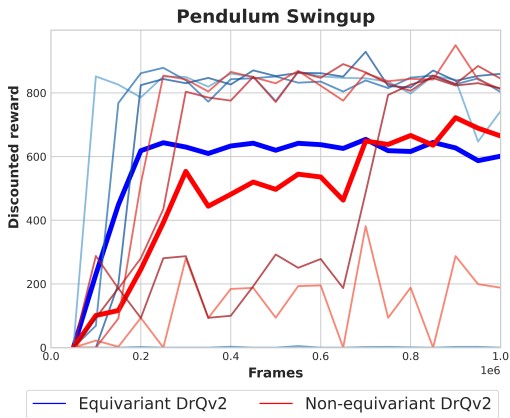

Figure 27: All runs of equivariant and non-equivariant DrQv2 on the DMC pendulum swingup task. Each method has 1 failed seed - the failed equivariant policy (blue) run is consistently near zero reward and the failed non-equivariant policy run (red) is around 200. Overall, the equivariant DrQv2 learns faster than the non-equivariant version when it succeeds.

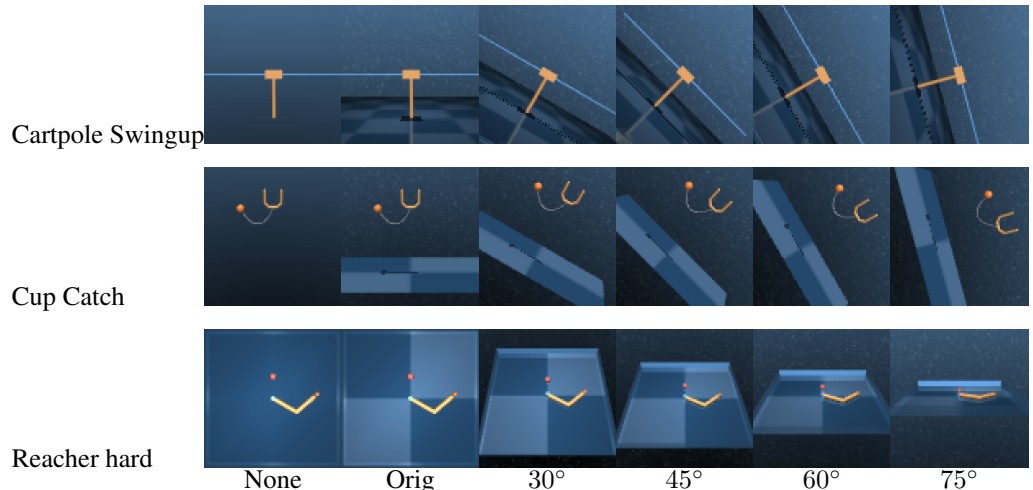

Table 3: Modifications to DMC domains for varying symmetry corruption levels. The gridded floor and background are removed to be fully equivariant (None) or the camera angle is modified to increase the level of symmetry corruption (roll for cartpole and cup catch, tilt for reacher).

## F.3 SEN ARCHITECTURE SEARCH

This section shows the architecture search for SEN. We consider three variations (all with similar amount of trainable parameters): 1) SEN conv (Figure 36): all layers are implemented using convolutional layers. 2) SEN fc1 (Figure 37) and SEN fc2 (Figure 20) replaces some layers in 1) with fully connected layers.

Figure 38 shows the comparison across the three variations. 'SEN fc2' shows the best performance.

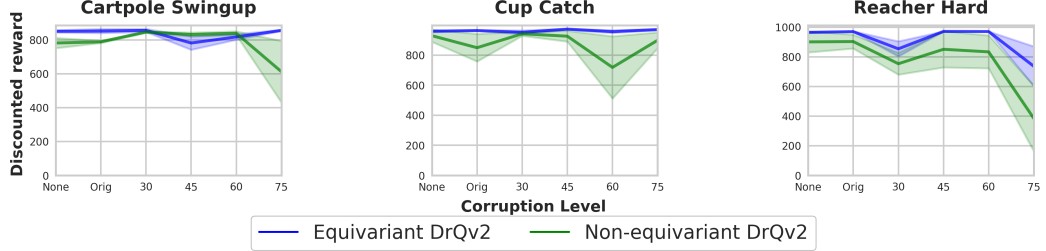

Figure 28: DMC performance comparison on various levels of symmetry corruptions. Both the equivariant and non-equivariant DrQv2 perform quite well even with increasing levels of corruption.

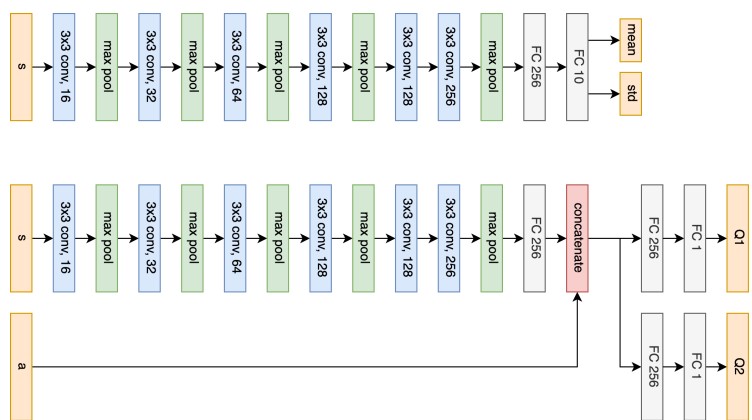

Figure 29: Network architecture of the 'fc1' variation for CNN SAC.

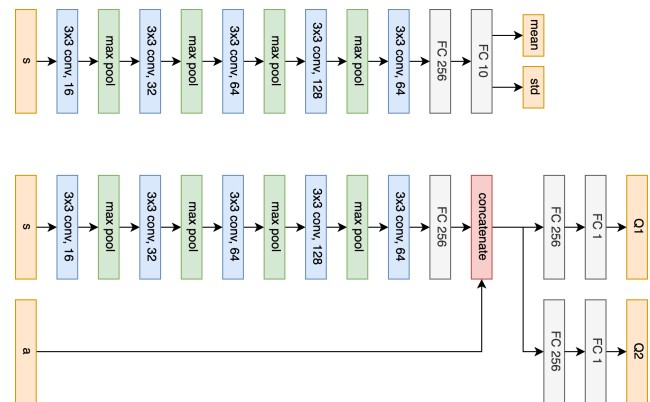

Figure 30: Network architecture of the 'fc2' variation for CNN SAC.

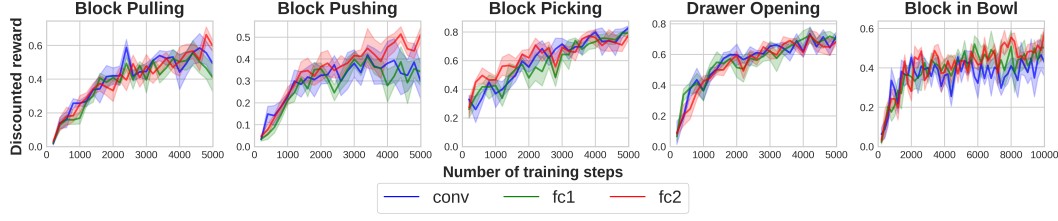

Figure 31: Architecture search for CNN SAC. The plots show the performance (in terms of discounted reward) of the evaluation policy. The evaluation is performed every 200 training steps. Results are averaged over four runs. Shading denotes standard error.

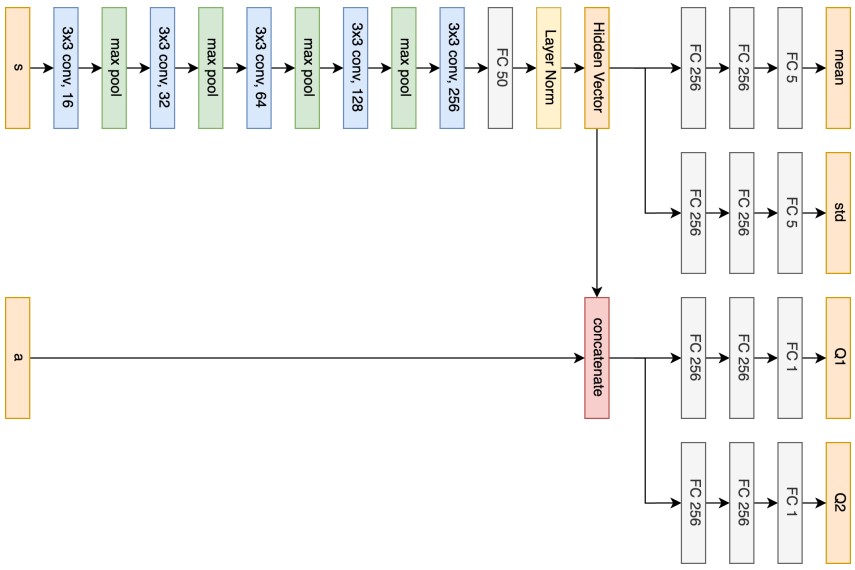

Figure 32: Network architecture of the 'sim enc' variation for FERM.

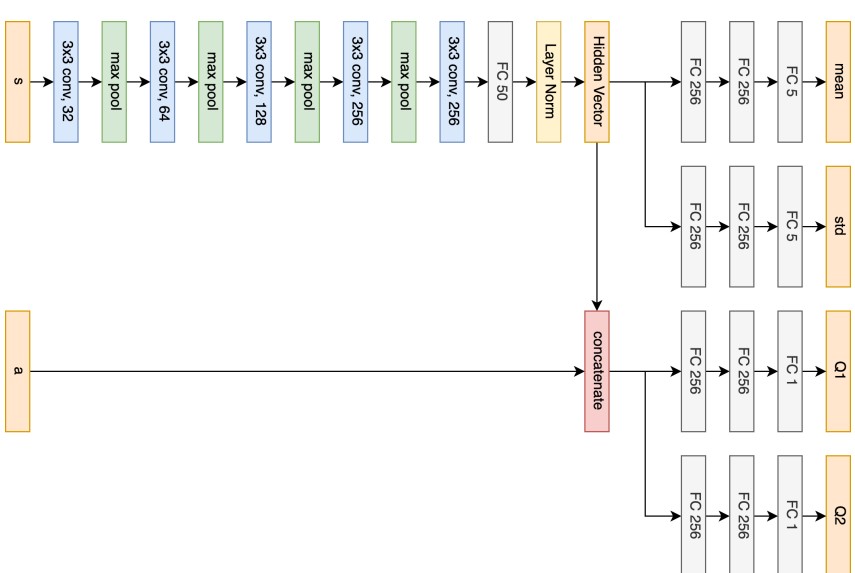

Figure 33: Network architecture of the 'sim total 1' variation for FERM.

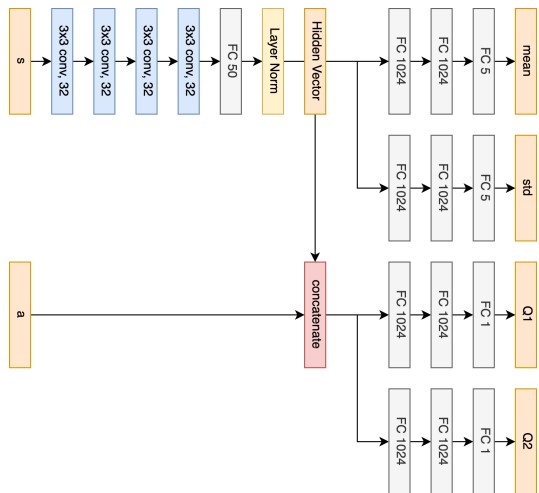

Figure 34: Network architecture of the 'ferm ori' variation for FERM.

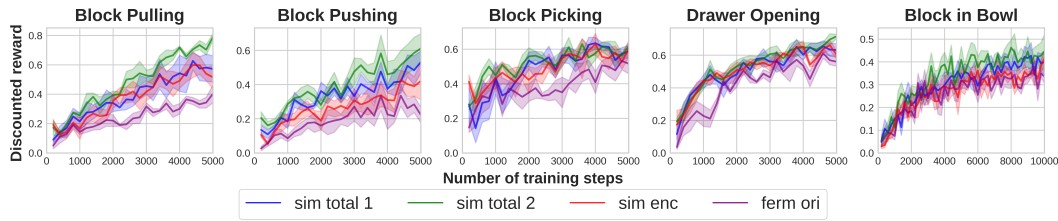

Figure 35: Architecture search for FERM. The plots show the performance (in terms of discounted reward) of the evaluation policy. The evaluation is performed every 200 training steps. Results are averaged over four runs. Shading denotes standard error.

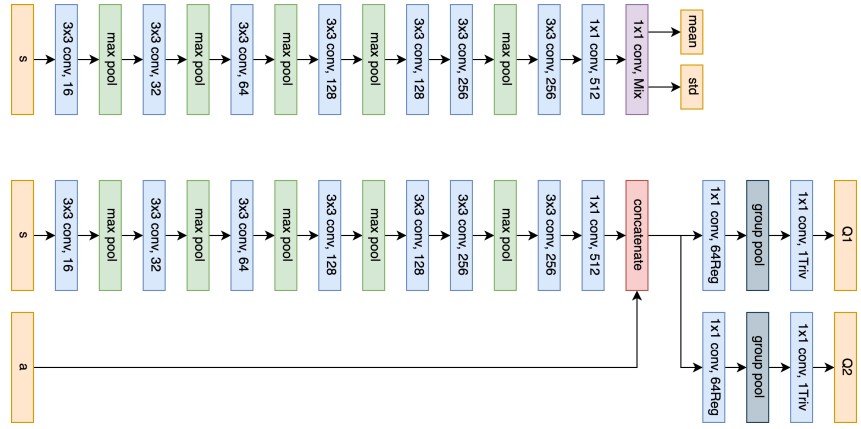

Figure 36: Network architecture of 'SEN conv' variation of SEN.

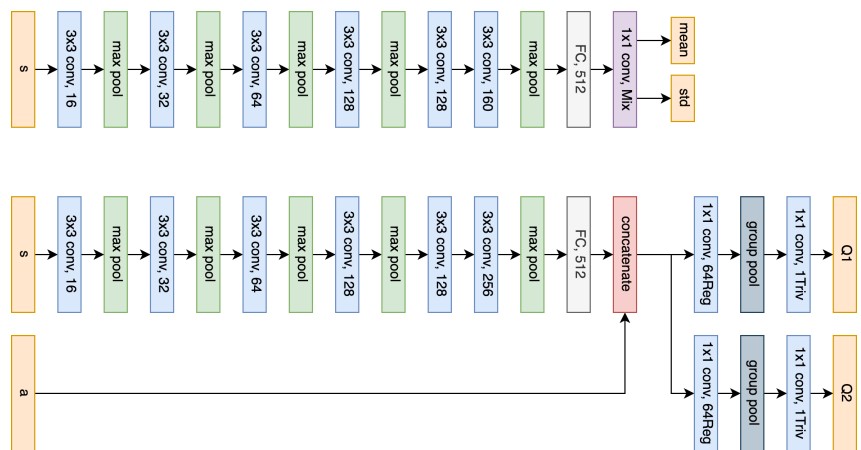

Figure 37: Network architecture of 'SEN fc1' variation of SEN.

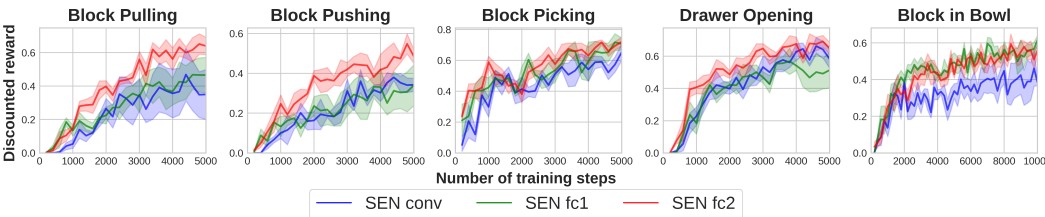

Figure 38: Architecture search for SEN. The plots show the performance (in terms of discounted reward) of the evaluation policy. The evaluation is performed every 200 training steps. Results are averaged over four runs. Shading denotes standard error.

