# OpenReview forum: "The Surprising Effectiveness of Equivariant Models in Domains with Latent Symmetry"
_ICLR.cc/2023/Conference — ICLR 2023 notable top 25%_

### Official Review · Reviewer_a4PJ · 2022-10-21

**Confidence:** 4
**Correctness:** 4
**Technical Novelty And Significance:** 3
**Empirical Novelty And Significance:** 3
**Recommendation:** 8

**Clarity, Quality, Novelty And Reproducibility:**

The clarity of the paper is great. I understood the authors' message without much effort, and I do not expect most of the audience to be challenged with that.

The quality of the work is also very good: the experiments are extensive and they prove the point empirically. The theoretical result makes sense: it is simple, elegant and useful.

I think the work is quite original, due to combining simple group theoretical arguments and strong experiments, and deserves its spot in the conference.

**Strength And Weaknesses:**

Strengths:

1. Very important idea that the equivariance may be latent, and not necessarily captured in the input transformation.

2. Useful definitions of correct, incorrect and extrinsic equivariances that capture the intuition of the problem.

3. Really neat proof of Proposition A.1.

Weaknesses:

1. The authors address it in the end of their paper, but it would be nice to consider the method in other tasks / settings. I wonder if the power of the equivariant policy would be even further demonstrated in a realistic scenario, where the states are actual natural images. In that setting the extrinsic equivariances will be "natural." Might be a cool experiment to try.

2. Some minor comments:

* It would be great if you could put Figure 2 on the first page to visualize what you mean in the discussion.

* In Section 4.1. it is not clear what the relationship between $\hat{\rho}_x$ and $\rho_x$ is. Can you give some examples for intuition early in the paper?

* Small typo in the proof of Proposition A.1. "labels and maximally" <- "labels are maximally".


**Summary Of The Paper:**

This paper defines three types of equivariance: correct, incorrect and extrinsic. The authors prove an upper bound on the accuracy when incorrect equivariance is used. Then the authors show empirically that using incorrect equivariance (e.g. using an equivariant neural network with incorrect labels) is worse than using a neural network, without the equivariant constraint. Finally, the authors shows that extrinsic equivariance can be helpful for policies in reinforcement learning. The authors experiment in two domains: 1) simple synthetic domain and 2) reinforcement learning tasks on robotic manipulation and DeepMind Control Suite.

**Summary Of The Review:**

I like the paper, because the problem of equivariance for neural networks is widely recognized, the proposed studies and theory are quite useful, and further research in that direction is promising, and may benefit from the findings in that paper.

---

> ### Author Response · Authors · 2022-11-16
> **Author Response to Reviewer a4PJ**
>
> The authors thank the reviewer for their helpful comments. Please see our comments below:
>
> > The authors address it in the end of their paper, but it would be nice to consider the method in other tasks / settings. I wonder if the power of the equivariant policy would be even further demonstrated in a realistic scenario, where the states are actual natural images. In that setting the extrinsic equivariances will be "natural." Might be a cool experiment to try.
>
> That’s a great point! We definitely agree the benefits of extrinsic equivariance are not limited to the cases we consider here. Trying this method on natural images would be a great experiment. We plan to extend the current approach and include more realistic data and a variety of different datasets in future work. We do point out that we mimic some realistic “effects” in the simulator, such as the ‘light effect’ corruption in Figure 3 and tilted observation in Figure 5.
>
> > It would be great if you could put Figure 2 on the first page to visualize what you mean in the discussion.
>
> Thanks for the suggestion, we have moved Figure 2 to the first page (now Figure 1 in the revision) and edited the image to better illustrate our core idea.
>
> > In Section 4.1. it is not clear what the relationship between $\hat{\rho}(x)$ and $\rho(x)$ is. Can you give some examples for intuition early in the paper?
>
> This is a good point, thanks for pointing this out. We added an example in Section 4.1.
>
> > Small typo in the proof of Proposition A.1. "labels and maximally" <- "labels are maximally".
>
> Thanks for pointing this out, we fixed this typo in the revision.

---

### Official Review · Reviewer_bQBQ · 2022-10-25

**Confidence:** 3
**Correctness:** 4
**Technical Novelty And Significance:** 4
**Empirical Novelty And Significance:** 4
**Recommendation:** 8

**Clarity, Quality, Novelty And Reproducibility:**

The paper is clear and well-written. For one minor note, definition 4.3 seems incomplete and is missing the equivariance definition.

**Strength And Weaknesses:**

Strengths
+ The experimental evaluation is extensive and significantly improves upon established results for equivariant SAC/RAD (Fig 7, 9, 11) and DrQv2 (Fig 12).
+ The appendix includes a significant amount of ablations and details, for example the tasks in Table 3 and Figure 27 on evaluating the DMC tasks with corruptions to the symmetry/camera pose

Weaknesses
+ While the paper is motivated by unknown latent symmetries, I thought it involve learning them. However, as reading the paper, I realized that the extrinsic equivariances still need to be manually specified as auxiliary equivariances that are simply other known transformations.

**Summary Of The Paper:**

This paper proposes to learn *extrinsic* symmetries and equivariances in settings that have latent symmetries that are challenging to manually impose with standard group equivariance. The method is evaluated mostly on pixel-based settings with convolutional networks with a significant focus on pixel reinforcement learning tasks.

**Summary Of The Review:**

I recommend to accept this paper as the idea of extrinsic equivariance is appealing and the experimental evaluation is convincing and improves upon many established results.

---

> ### Author Response · Authors · 2022-11-16
> **Author Response to Reviewer bQBQ**
>
> The authors thank the reviewer for their thoughtful review. Please see our comments below:
>
> > While the paper is motivated by unknown latent symmetries, I thought it involve learning them. However, as reading the paper, I realized that the extrinsic equivariances still need to be manually specified as auxiliary equivariances that are simply other known transformations.
>
> You are correct, the extrinsic equivariance needs to be manually specified. As we point out in the introduction, the problem we are trying to address is when ‘the designer knows that a latent symmetry is present in the problem but cannot easily express how that symmetry acts in the input space’. This may be the case in many scenarios. For example, if a pair of images represent a 2D rotation of an object, the group action may be difficult to characterize if the input image is not a top-down view and has a skewed perspective. In this scenario, one can easily identify the symmetry group (SO(2)) but the group action on the input space is not straightforward to compute. We have added some sentences in the related work section to clarify the difference between our work and other symmetry learning works (There is a difference between the latent symmetry group and the group action on input space).
>
> > definition 4.3 seems incomplete and is missing the equivariance definition.
>
> In extrinsic equivariance (definition 4.3), for $x\in D$, $\hat{\rho}_x(g)x$ will be out of the input space, thus $f(\hat{\rho}_x(g)x)$ (i.e., the ground truth label) is undefined, so that the equivariance definition with respect to the ground truth function does not apply here. As a side note, the equivariance with respect to the model function $f_\phi(\hat{\rho}_x(g)x) = \rho_y(g)f_\phi(x)$ does apply to all three definitions, as stated in Section 4.1.

---

### Official Review · Reviewer_6UwC · 2022-10-27

**Confidence:** 3
**Correctness:** 4
**Technical Novelty And Significance:** 3
**Empirical Novelty And Significance:** 2
**Recommendation:** 8

**Clarity, Quality, Novelty And Reproducibility:**

The paper is very well-written and of high quality. I found it easy to follow and the experiments were well-chosen for demonstrating the different symmetries. The Appendix was detailed. I browsed the including software for the experiments but did not run them. I am reasonably confident that the work is reproducible. The work is novel, but it can feel a bit incremental as it's main contribution isn't a new model.

**Strength And Weaknesses:**

 Overall, I view the paper as a useful contribution to the literature
 as it provides greater clarity for where equivariant models can really
 shine. While the paper does introduce an equivariant version of DrQv2,
 I feel the contribution is introducing the distinction between "incorrect
 equivariance" and "extrinsic equivariance".

 The main weakness of the paper is it doesn't introduce new models. All
 the theory and most of the experiments are done using existing equivariant
 models in the literature. Much of the benefits of equivariant models are
 already known as the related work highlights and there are already many
 methods for learning latent symmetries.

**Summary Of The Paper:**

 This paper explores the settings where an equivariant model can
 improve sampling efficiency and generalization by proposing a
 distinction between equivariant models that preserve the true latent
 symmetry and those which make it impossible to represent. These
 equivariant models that preserve the true latent symmetry even while
 not explicitly modeling it are called "extrinsic
 equivariance". Through a combination of theoretical and empirical
 results, this paper shows how extrinsic symmetries improve models. In
 fact, some of the improved generalization may come from
 "extrinsically equivariant" models transforming input data out of
 distribution in a way that non-equivariant models simply don't do.


**Summary Of The Review:**

A great contribution to the equivariant models literature, but the value is not in any one particular method.

---

> ### Author Response · Authors · 2022-11-16
> **Author Response to Reviewer 6UwC**
>
> The authors thank the reviewer for their careful review. Please see our comments below:
>
> > The main weakness of the paper is it doesn't introduce new methods. All the theory and most of the experiments are done using existing equivariant methods in the literature.
>
> The main contribution of our work is not a new **model** (though we do propose a novel equivariant DrQv2 model, as the reviewer points out), however, we believe our proposal to repurpose fully symmetric models for non-fully symmetric problems does constitute a new **method**.
>
> > Much of the benefits of equivariant models are already known as the related work highlights
>
> It is true that many prior works have shown the benefits of equivariant models, while they use equivariant models under the ‘correct equivariance’ scenario. We believe that our work opens the gate for the equivariant models to be useful for a wider area of applications.
>
> > and there are already many methods for learning latent symmetries.
>
> Our work can also be considered as learning latent symmetries, but compared to prior works, our approach does not require a carefully designed network architecture. We also compare with the Symmetric Embedding Network (Park et al., 2022), one very new work in learning latent symmetries, in Figure 7 and show that our work outperforms it.

---

> > ### Comment · Reviewer_6UwC · 2022-11-19
> > **Re: Author response**
> >
> > Thanks for the clarifications. I'll adjust my review to be more precise about the contributions of the paper.

---

### Official Review · Reviewer_ZXUL · 2022-10-29

**Confidence:** 2
**Correctness:** 4
**Technical Novelty And Significance:** 3
**Empirical Novelty And Significance:** 3
**Recommendation:** 8

**Clarity, Quality, Novelty And Reproducibility:**

I consider the work novel. However, as pointed out in the weaknesses part, there are still rooms to improve clarity.

**Strength And Weaknesses:**

Strength
* This work empirically shows that extrinsic equivariance can still aid learning, which is very well-motivated contribution to model architecture with image transformation augmentation.
* Empirical evaluations are strong and comprehensive, covering many kinds of transformations, supervised learning, and reinforcement learning.

Weaknesses
* The scale of the model and data used in these experiments are still considered small. Given the results in slightly larger data-regime, it's not clear how much it could be useful with large model and large data.
* When the term correct, incorrect, and extrinsic equivariance/symmetry appear the first time in the paper, they are not well-explained. Even in the following, it's still unclear what the definition is and examples are somewhat unclear. I wonder if it would be useful to at least have section in appendix to clarify is the main paper space is too constrained.


**Summary Of The Paper:**

This work defines correct equivariance, incorrect equivariance, and extrinsic equivariance. Correct symmetry means that the model symmetry correctly reflects a symmetry present in the ground truth function, for which it is correct to enforce equivariance constraints. Extrinsic equivariance is when the equivariant constraint in the equivariant network enforces equivariance to out-of-distribution data. The authors theoretically demonstrate the upper bound performance for an incorrectly constrained equivariant model. In a supervised classification setting, they empirically show that a model with extrinsic equivariance can aid learning compared with an unconstrained model, especially in a low-data regime. Then, they explore this idea in a RL context and show that an extrinsically constrained model can outperform state-of-the-art conventional CNN baselines.

**Summary Of The Review:**

I lean to acceptance but would like the authors to improve the clarify and comment on effectiveness in large-data regime.

---

> ### Author Response · Authors · 2022-11-16
> **Author Response to Reviewer ZXUL**
>
> The authors thank the reviewer for their insightful review. Please see our comments below:
> > The scale of the model and data used in these experiments are still considered small. Given the results in slightly larger data-regime, it's not clear how much it could be useful with large model and large data.
>
> A significant benefit of equivariance is to improve sample efficiency by enabling the model to learn faster with less data [A]. This is particularly important in robotics where acquiring samples is expensive. Thus our manipulation experiments focus on the low-data regime to highlight the benefits of equivariance, even when it is extrinsic. However, we do also consider larger-data regimes as in the DMC experiments in Figure 12 which use larger models and data where it uses 1e6~3e6 samples (at least 2 orders of magnitude more data than robotic manipulation). Figure 12 shows that extrinsic equivariance still noticeably learns faster. This suggests that extrinsic equivariance would still be useful with larger models and more data and we will investigate this further in future work.
>
> [A] Cohen, T.S., “Equivariant convolutional networks”, 2021.
>
> > When the term correct, incorrect, and extrinsic equivariance/symmetry appear the first time in the paper, they are not well-explained. Even in the following, it's still unclear what the definition is and examples are somewhat unclear. I wonder if it would be useful to at least have section in appendix to clarify is the main paper space is too constrained.
>
> Thanks for pointing this out, we added a brief explanation of correct, incorrect, and extrinsic equivariance in the introduction. We also added a new section in the Appendix (Appendix B in the revision) to extend Section 4.2 with detailed examples to better illustrate those terms.

---

> > ### Comment · Reviewer_ZXUL · 2022-11-26
> > **Thanks for the response**
> >
> > Thanks for the clarification. I still consider all the experiments small-scale in today's standard. That includes DMC, which is single-task and with limited data distribution. I still suspect the benefit of the proposed approach will potentially decay with much more data and larger models. But I recognize the benefits of the proposed approach in the regime of little in-domain data. I will increase my rating.

---

### Author Response · Authors · 2022-11-16
**Summary of Revision**

The authors thank all reviewers for their careful and helpful review. We appreciate that all reviewers like our idea/definition of extrinsic equivariance and acknowledge our experimental results. We have edited the paper to incorporate reviewer feedback and believe it has improved as a result (edits labeled in blue in the manuscript). Here is a summary of the revisions:
1. Moved Figure 2 in the initial submission to the first page (now Figure 1) and edited it to better illustrate the main idea of the paper.
2. Added a high-level explanation of correct, incorrect, and extrinsic equivariance in the introduction where they first appear.
3. Edited the related work section on ‘Symmetric Representation Learning’ to better illustrate the difference between our work and the prior works.
4. Added an example to illustrate the relationship between $\hat{\rho}(x)$ and $\rho(x)$ in Section 4.1.
5. Fixed a typo in Appendix A.
6. Added examples to better explain correct, incorrect, and extrinsic equivariance in Appendix B.

---

> ### Author Response · Authors · 2022-11-18
> **Supplementary Video**
>
> Dear reviewers,
>
> We have uploaded a supplementary video to help explain our idea. You can find the video through this link https://www.youtube.com/watch?v=V_448CfUOwE (uploaded using an anonymous account) or in the 'Supplementary Material' zip file.

---

### Decision · Program_Chairs · 2023-01-20

**Decision:**

Accept: notable-top-25%

**Justification For Why Not Higher Score:**

Paper only includes relatively small experiments.

**Justification For Why Not Lower Score:**

The paper introduces some interesting new concepts that will be of interest to the equivariant networks community.

**Metareview: Summary, Strengths And Weaknesses:**

This paper introduces the concepts of correct/incorrect/extrinsic equivariance and shows that while incorrect equivariance limits performance, extrinsic equivariance can be of benefit. The reviewers unanimously rated the paper 8, and found it to be well written. The reviewers appreciated the new concepts introduced in the paper, as well as the comprehensive experiments involving different kinds of transformations in supervised and reinforcement learning. The main limitations that were discussed are that the experiments involved relatively small models and datasets, and that the paper does not introduce a new equivariant model (although it could be argued that it introduces a new method). Overall this is a solid paper and I recommend that it be accepted to ICLR.

**Note From Pc:**

if the above contains the word "oral" or "spotlight" please see: "oral" presentation means -> notable-top-5% and "spotlight" means -> notable-top-25%. As stated in our emails, we are disassociating presentation type from AC recommendations